# How to Upscale Neural Networks with Scaling Law?

**Ayan Sengupta**[*,◇]                                                        *ayan.sengupta@ee.iitd.ac.in*
*Indian Institute of Technology Delhi*

**Yash Goel**[*]                                                                  *ee1210984@ee.iitd.ac.in*
*Indian Institute of Technology Delhi*

**Tanmoy Chakraborty**                                                    *tanchak@ee.iitd.ac.in*
*Indian Institute of Technology Delhi*

**Reviewed on OpenReview:** *https://openreview.net/forum?id=AL7NOUOfgI*

## Abstract

Neural scaling laws have revolutionized the design and optimization of large-scale AI models by revealing predictable relationships between model size, dataset volume, and computational resources. Early research established power-law relationships in model performance, leading to compute-optimal scaling strategies. However, recent studies highlighted their limitations across architectures, modalities, and deployment contexts. Sparse models, mixture-of-experts, retrieval-augmented learning, and multimodal models often deviate from traditional scaling patterns. Moreover, scaling behaviors vary across domains such as vision, reinforcement learning, and fine-tuning, underscoring the need for more nuanced approaches. In this survey, we synthesize insights from current studies, examining the theoretical foundations, empirical findings, and practical implications of scaling laws. We also explore key challenges, including data efficiency, inference scaling, and architecture-specific constraints, advocating for adaptive scaling strategies tailored to real-world applications. We suggest that while scaling laws provide a useful guide, they do not always generalize across all architectures and training strategies.

## 1 Introduction

Scaling laws have become a fundamental aspect of modern AI development, especially for large language models (LLMs). In recent years, researchers have identified consistent relationships between model size, dataset volume, and computational resources, demonstrating that increasing these factors leads to systematic improvements in performance. These empirical patterns have been formalized into mathematical principles, known as *scaling laws*, which provide a framework for understanding how the capabilities of neural networks evolve as they grow. Mastering these laws is crucial for building more powerful AI models, optimizing efficiency, reducing costs, and improving generalization.

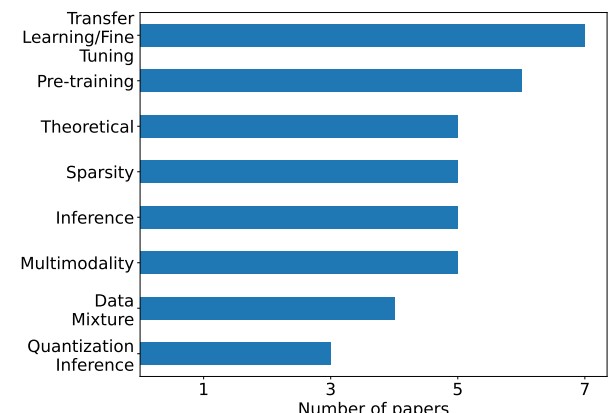

Figure 1: Papers surveyed under different categories. A detailed list of papers is provided in Table 13 of Appendix A.

| Category | Choshen et al. (2024) | Li et al. (2024b) | Ours |
|---|---|---|---|
| Covers neural scaling laws broadly | Yes | No | Yes |
| Discusses fitting methodologies | Yes | Yes | Yes |
| Analyzes architectural considerations | No | Limited | Yes |
| Includes data scaling and pruning | No | Limited | Yes |
| Explores inference scaling | No | Limited | Yes |
| Considers domain-specific scaling | No | No | Yes |
| Provides practical guidelines | Yes | Yes | Yes |
| Critiques limitations of scaling laws | Limited | Yes | Yes |
| Proposes future research directions | Limited | Yes | Yes |

Table 1: Key differences between our survey and existing surveys on neural scaling laws (Choshen et al., 2024; Li et al., 2024b).

The study of neural scaling laws gained prominence with the foundational work of Kaplan et al. (2020), who demonstrated that model performance follows a power-law relationship with respect to size, data, and compute. Their findings suggested that larger language models (LMs) achieve lower loss when trained on sufficiently large datasets with increased computational resources. Later, Hoffmann et al. (2022) refined these ideas, introducing the notion of compute-optimal scaling, which revealed that training a moderate-sized model on a larger dataset is often more effective than scaling model size alone. However, recent studies (Muennighoff et al., 2023; Caballero et al., 2023; Krajewski et al., 2024) have challenged the universality of these laws, highlighting cases where sparse models, mixture-of-experts architectures, and retrieval-augmented methods introduce deviations from traditional scaling patterns. These findings suggested that while scaling laws provide a useful guide, they do not always generalize across all architectures and training strategies.

Despite the growing importance of scaling laws, existing research remains fragmented, with limited synthesis of theoretical foundations, empirical findings, and practical implications. Given the rapid evolution of this field, there is a need for a structured analysis that consolidates key insights, identifies limitations, and outlines future research directions. While theoretical studies have established the mathematical principles governing scaling, their real-world applications, such as efficient model training, optimized resource allocation, and improved inference strategies, are less explored. To address this gap, we reviewed over 50 research articles (Figure 1 highlights papers on scaling laws on different topics) to comprehensively analyze scaling laws, examining their validity across different domains and architectures.

While prior surveys have made valuable contributions to understanding scaling laws, they have primarily focused on specific aspects of the scaling phenomenon (See Table 1). Choshen et al. (2024) emphasized statistical best practices for estimating and interpreting scaling laws using training data, while Li et al. (2024b) emphasized on methodological inconsistencies and reproduction crisis in existing scaling laws. Our survey distinguishes itself by offering comprehensive coverage of architectural considerations, data scaling implications, and inference scaling – areas that previous surveys either overlooked or addressed only partially.

## 2    Taxonomy of neural scaling laws

Understanding the scaling laws of neural models is crucial for optimizing performance across different domains. We predominantly explore the scaling principles for language models, extending to other modalities such as vision and multimodal learning. We also examine scaling behaviors in domain adaptation, inference, efficient model architectures, and data utilization. We highlight the taxonomy tree of scaling laws research in Figure 2 with list of paper covered under each taxonomy branch in Table 2. As highlighted in Figure 1, neural scaling laws have been proposed predominantly for pre-training and fine-tuning scaling of large neural models. Among the models studied, as highlighted in Figure 3a, decoder-only Transformers dominate the subject, followed by vision transformers (ViT) and Mixture-of-Experts (MoE).

---

*Equal contribution
◇Corresponding author

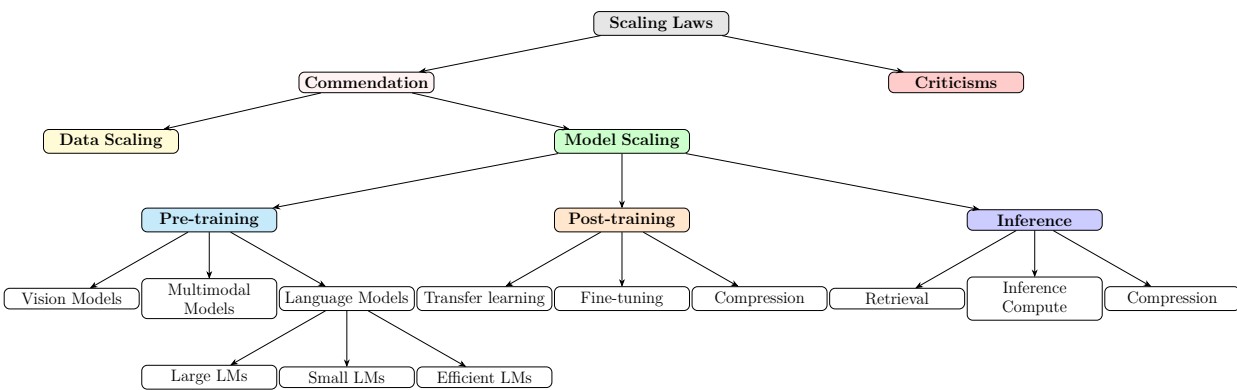

Figure 2: A taxonomy of neural scaling laws.

| Taxonomy | Cited Papers |
|---|---|
| Data Scaling | Muennighoff et al. (2023); Ye et al. (2024); Liu et al. (2024); Kang et al. (2024); Que et al. (2024); Allen-Zhu & Li (2024); Tao et al. (2024) |
| Pre-training → Vision Models | Zhai et al. (2022); Alabdulmohsin et al. (2022) |
| Pre-training → Multimodal Models | Henighan et al. (2020); Aghajanyan et al. (2023); Li et al. (2024a) |
| Pre-training → Language Models → Large LMs | Kaplan et al. (2020); Hoffmann et al. (2022); Tay et al. (2022); Caballero et al. (2023) |
| Pre-training → Language Models → Small LMs | Hu et al. (2024) |
| Pre-training → Language Models → Efficient LMs | Clark et al. (2022); Krajewski et al. (2024); Yun et al. (2024) |
| Post-training → Transfer learning | Hernandez et al. (2021) |
| Post-training → Fine-tuning | Zhang et al. (2024); Chen et al. (2024c); Lin et al. (2024b) |
| Post-training → Model Compression | Frantar et al. (2023); Chen et al. (2024b) |
| Inference → Retrieval | Shao et al. (2024) |
| Inference → Inference Compute | Brown et al. (2024); Wu et al. (2024); Sardana et al. (2024) |
| Inference → Compression | Dettmers & Zettlemoyer (2023); Cao et al. (2024); Kumar et al. (2024) |
| Criticisms | Sorscher et al. (2023); Diaz & Madaio (2024) |

Table 2: Papers covered under different taxonomy.

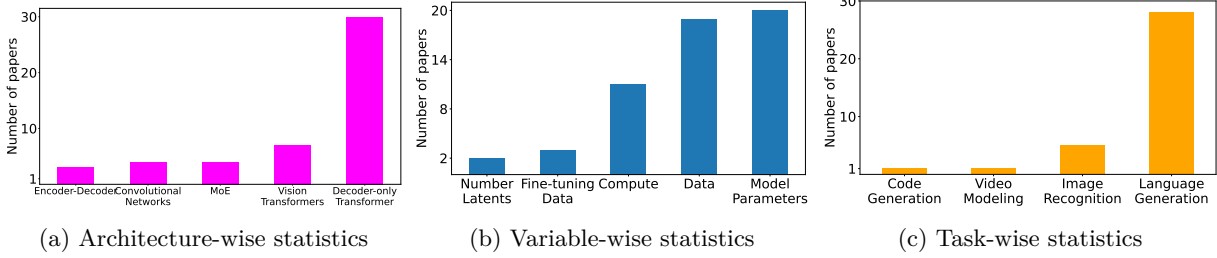

(a) Architecture-wise statistics  (b) Variable-wise statistics  (c) Task-wise statistics

Figure 3: Number of papers studied in this survey for different model architectures (a), scaling variables (b), and scaling tasks (c). The detailed paper list is provided in Table 13 of Appendix A.

The most common neural scaling laws take the form of power laws (Equation 1), where the model's loss ($L$) or performance metric assumes to follow a predictable relationship with different scaling variables,

$$L(P_{i....n}) = \sum_{i=1}^{n} \alpha_i \cdot P_i^{-\beta_i} \tag{1}$$

with appropriate scaling parameters $\beta_i$ and fitting parameters $\alpha_i$ for different scaling parameter $P_i$. Figure 3b highlights that the number of model parameters and data size are the most common used scaling factors. Intuitively, the exponent $\beta_i$ measures the rate of performance saturation: larger $\beta_i$ implies faster improvement with scale, while smaller $\beta_i$ indicates slower convergence. The empirical intuition of power laws in neural models stems from statistical physics and information theory principles, where performance improvements exhibit diminishing returns as scale increases. In neural networks, the marginal gain in loss reduction with respect to an increase in model parameters or data size tends to follow a sublinear pattern, indicating similar behaviour across scales. This observation aligns with complexity–capacity trade-offs, where each additional parameter contributes progressively less new information. As shown by Kaplan et al. (2020) and later formalized in Hoffmann et al. (2022), the power-law exponents quantify how efficiently a model converts compute and data into predictive performance. The exact forms of all the scaling laws are highlighted in Table 14 of Appendix A.

Among all the tasks, Figure 3c suggests that language generation is the most common task used for developing these scaling laws, where the training cross-entropy loss is widely used to fit the laws. Based on the values obtained empirically, the scaling laws are fitted with non-linear optimization, most commonly by running algorithms like least square and BFGS (Broyden-Fletcher-Goldfarb-Shanno). Statistical methods like goodness-of-fit metrics are used to validate the correctness of the fitted curves. We elaborate on the evaluation of neural scaling laws in Section 3. In the following sections, we review the existing literature on neural scaling across various domains.

**Model scaling** includes both parameter and data scaling. Parameter scaling is often studied in decoder-only Transformers (Kaplan et al., 2020; Hoffmann et al., 2022), with newer works addressing small and efficient models (Hu et al., 2024; Clark et al., 2022). These studies establish power-law relationships between loss and model size or compute (Equation 1). In parallel, power laws also implicitly encode the compute-optimal frontier: the point where increasing parameters or data yields equivalent marginal benefits. This equilibrium condition forms the basis of the Chinchilla scaling rule, where $D \propto N$, derived by equating the partial derivatives of Equation 1 with respect to $N$ and $D$.

**Data scaling** research has proposed laws for optimizing mixtures (Ye et al., 2024), repeated training exposures (Muennighoff et al., 2023), vocabulary size (Tao et al., 2024), and knowledge capacity (Allen-Zhu & Li, 2024). These formulations often assume that information gain from additional data follows a diminishing-return process similar to Zipfian or Pareto distributions observed in natural language frequency, justifying why scaling laws adopt a power-law rather than logarithmic form.

**Pre-training scaling laws** extend beyond language to vision and multimodal settings. Vision models exhibit power-law scaling that saturates at large compute (Zhai et al., 2022), while multimodal models demonstrate competition-to-synergy transitions as scale increases (Aghajanyan et al., 2023).

**Post-training scaling** captures fine-tuning and transfer learning behaviors. Transfer scaling shows larger pre-trained models yield better generalization with limited downstream data (Hernandez et al., 2021). Recent works propose scaling laws for PEFT (Zhang et al., 2024), downstream loss prediction (Chen et al., 2024c), and early stopping (Lin et al., 2024b).

**Inference scaling** explores compute-efficient strategies during model deployment. Adaptive test-time compute (Chen et al., 2024a; Brown et al., 2024) and retrieval augmentation (Shao et al., 2024) allow small models to rival larger ones. Inference-specific scaling laws characterize the tradeoff between sampling cost and performance (Wu et al., 2024).

**Efficient model scaling** addresses sparsity, quantization, and distillation. Sparse and MoE models provide multiplicative efficiency gains (Krajewski et al., 2024), while pruning and quantization laws enable compute-aware compression (Chen et al., 2024b; Cao et al., 2024).

**Scaling behavior in reinforcement learning** (RL) diverges from language or vision tasks. In single-agent RL, performance scales sublinearly with model size and environment interaction (Hilton et al., 2023). Horizon length, rather than task difficulty, determines scaling efficiency. In multi-agent games, predictable scaling laws govern compute-to-performance relationships, but generalization to complex domains like Chess or Go remains limited (Neumann & Gros, 2023). Meanwhile, graph neural networks (GNNs) lack stable

| Taxonomy Node | Addressed RQs |
|---|---|
| Model scaling | RQ1, RQ2, RQ8 |
| Data scaling | RQ3 |
| Post-training scaling | RQ5 |
| Inference scaling | RQ4 |
| Efficient and compressed model scaling | RQ6, RQ7 |

Table 3: Mapping taxonomy categories to relevant research questions.

scaling laws; despite self-supervised loss improving with more data, downstream performance often fluctuates unpredictably (Ma et al., 2024).

Finally, the taxonomy captures two outer branches: **commendations**, such as practical data laws and compression-aware training (Liu et al., 2024), and **criticisms**, which question the generalizability and reproducibility of scaling laws (Sorscher et al., 2023; Diaz & Madaio, 2024). Detailed discussion on these scaling law studies are provided in Appendix A.

## 3 Fitting and validating scaling laws

Fitting scaling laws involves several key methodological choices that can significantly impact the final results and conclusions. The choice of optimization approach, loss function, initialization strategy, and validation method all play crucial roles in determining the reliability and reproducibility of scaling law studies.

### 3.1 Optimization methods

The most common approaches for fitting scaling laws involve non-linear optimization algorithms like BFGS (Broyden-Fletcher-Goldfarb-Shanno) (used by Frantar et al. (2023)), L-BFGS (used by Tao et al. (2024)) and least squares (used by Caballero et al. (2023)). Some studies (Covert et al., 2024; Hashimoto, 2021) also use optimizers like Adam or Adagrad, though these may be less suitable for scaling law optimization due to their data-hungry nature and assumptions about gradient distributions. Recent works (Hoffmann et al., 2022; Sorscher et al., 2023; Yun et al., 2024) emphasize that the choice of optimization method directly affects the stability of fitted exponents and intercepts. Second-order methods such as BFGS and L-BFGS typically converge more reliably for low-dimensional, smooth objective surfaces, whereas first-order methods can oscillate under noisy residuals.

### 3.2 Loss functions and objectives

Several loss functions are commonly used for fitting scaling laws:

- **Mean squared error (MSE)**: Emphasizes larger errors due to quadratic scaling (used by Ghorbani et al. (2021)).
- **Mean absolute error (MAE)**: Provides more robust fitting less sensitive to outliers (used by Hilton et al. (2023)).
- **Huber loss**: Combines MSE's sensitivity to small errors with MAE's robustness to outliers (used by Hoffmann et al. (2022)).

While most studies minimize residuals between predicted and observed loss, the choice of loss function carries distinct theoretical implications. MSE magnifies the influence of outliers, which can distort estimated power-law exponents, whereas Huber loss mitigates this by switching to a linear penalty beyond a threshold. Hoffmann et al. (2022) used Huber loss to down-weight outlier data points in the Chinchilla law, improving stability in under-trained regimes. In contrast, Muennighoff et al. (2023) and Sorscher et al. (2023) employed MSE on log-transformed loss, prioritizing proportional (relative) rather than absolute error.

### 3.3 Initialization strategies

The initialization of scaling law parameters proves to be critically important for achieving good fits. Common approaches include grid search over parameter spaces (Aghajanyan et al., 2023), random sampling from parameter ranges (Frantar et al., 2023), and multiple random restarts to avoid local optima (Caballero et al., 2023). Caballero et al. (2023); Li et al. (2024b) show that initialization sensitivity increases for broken or multi-regime scaling laws. Caballero et al. (2023) used multiple initializations per run to stabilize piecewise fits, while Aghajanyan et al. (2023) proposed grid-based seeding for cross-modal scaling consistency. Proper initialization ensures convergence to the global rather than local optima, crucial for robust exponent estimation.

### 3.4 Validation methods

It is hugely important to understand if the scaling law fit achieved is accurate and valid. Most of the papers surveyed lack in validating their fits. Several approaches can help validating the effectiveness of scaling law fits. Statistical methods like computing confidence intervals can act as a goodness-of-fit metric (Alabdulmohsin et al., 2022). Furthermore, researchers can perform out-of-sample testing by extrapolation to larger scales (Hoffmann et al., 2022). Yun et al. (2024); Covert et al. (2024) additionally use RMSE-based model selection, likelihood-based comparisons, and extrapolation validation. Covert et al. (2024) applied maximum likelihood estimation to derive uncertainty intervals for fitted parameters, while Yun et al. (2024) validated on held-out extrapolated data points. Extrapolation testing remains the gold standard for evaluating scaling law robustness, whether the law generalizes beyond observed scales.

### 3.5 Limitations of fitting techniques

Li et al. (2024b) revealed several critical methodological considerations in fitting scaling laws. Different optimizers can converge to notably different solutions even with similar initializations, underscoring the need for careful justification of optimizer choice. Similarly, the analysis showed that different loss functions can produce substantially different fits when working with real-world data containing noise or outliers, suggesting that loss function selection should be guided by specific data characteristics and desired fit properties. Perhaps most importantly, the paper demonstrated that initialization can dramatically impact the final fit, with some methods exhibiting high sensitivity to initial conditions. Furthermore, many fits implicitly assume single-regime power laws, overlooking emerging evidence for broken or multi-phase scaling (Caballero et al., 2023). Systematic reporting of residual distributions, uncertainty intervals, and robustness checks is now regarded as best practice for reproducibility. Together, these findings emphasize the importance of thorough methodology documentation across all aspects of the fitting process - from optimizer selection and loss function choice to initialization strategy - to ensure reproducibility and reliability in scaling law studies.

In the next section, we formulate key research questions (mapping between the taxonomy and research questions highlighted in Table 3) derived from these studies and present practical guidelines for leveraging scaling laws in real-world model development.

## 4 Research questions and guidelines

Grounded in the taxonomy of neural scaling laws (Figure 2), we identify key research questions spanning six dimensions: *model scaling*, *architectural bottlenecks*, *inference scaling*, *data scaling*, *post-training strategies*, and *efficient model design*. For each, we synthesize multiple studies to extract overarching patterns, identify conflicting evidence, and propose actionable guidelines for researchers and practitioners navigating large-scale model development.

**RQ1. Importance on model and pre-training data size on performance [taxonomy: model scaling → pre-training]**

Kaplan et al. (2020) established a power-law relationship:

$$L(N, D) = \left[ \left( \frac{N_c}{N} \right)^{\frac{\alpha_N}{\alpha_D}} + \frac{D_c}{D} \right]^{\alpha_D}, \quad D \propto N^{0.74}. \tag{2}$$

Hoffmann et al. (2022) refined this into a compute-optimal formulation:

$$L(N, D) = \frac{A}{N^\alpha} + \frac{B}{D^\beta} + E, \quad D \propto N. \tag{3}$$

Recent research has challenged linear extrapolations. Muennighoff et al. (2023) and Sardana et al. (2024) showed that training small models longer can outperform larger models, especially under constrained data. Caballero et al. (2023) proposed Broken Neural Scaling Laws (BNSL):

$$L(N, D) = \begin{cases} aN^{-\alpha} + bD^{-\beta}, & N < N_c \\ cN^{-\alpha'} + dD^{-\beta'}, & N \geq N_c \end{cases} \tag{4}$$

**Where these laws work (and fail).** As summarized in Table 4, Kaplan-style laws are effective for large, dense autoregressive LMs trained on abundant, i.i.d. text with full convergence, but they *break down* when data is exhausted, models are under-trained, or regimes depart from i.i.d. text. Chinchilla's compute-optimal form works well when both model and data are co-optimized on curated corpora, yet it can fail in multimodal or highly redundant-data regimes and assumes uniform data quality and linear cost. BNSL accurately models performance when clear saturation or "phase changes" occur across scale, but it is sensitive to breakpoint initialization, noise in small-scale regions, and can mislead when extrapolating far beyond the fitted regimes. Figure 4 demonstrates an instance of "double-descent"/multi-regime behavior (Nakkiran et al., 2021) where a single-slope Kaplan fit systematically misses the curvature, while a BNSL fit, with learned breakpoints (annotated near ∼175 and ∼450 training examples in the figure), captures the transition and yields non-negligible lower extrapolation error (RMSE of 0.02 vs. 0.14). This example highlights why piecewise formulations are needed once the learning curve crosses saturation or phase-change regions.

> **Synthesis and guidelines**
>
> - Model scaling success depends not only on size but also on training strategy, data quality, and saturation thresholds.
> - Use Table 4 to match regime to law: Kaplan/Chinchilla for dense, well-curated i.i.d. text at scale; BNSL when empirical curves show slope changes or saturation; and data-aware/repetition-aware formulations in data-limited settings.
> - Before extrapolating, test for regime breaks (as in Figure 4); if present, prefer piecewise fits and report breakpoint uncertainty to avoid overconfident forecasts.
> - Practitioners should allocate compute across parameters, data, and training duration based on observed inflection points. Use Kaplan/Chinchilla scaling when data is abundant; otherwise, extend training epochs or adopt data-efficient curricula (see Figure 7a).

**RQ2. Scaling behaviors for different neural architectures [taxonomy: model scaling → pre-training → architecture]**

According to Tay et al. (2022), the vanilla Transformer consistently demonstrates superior scaling properties ($P \propto C^\alpha$, where $P$ is the performance metric, $C$ represents compute, and $\alpha$ are fitting parameters) compared to other architectures, even though alternative designs might perform better at specific sizes. Architectural bottlenecks manifest differently across these designs. For instance, linear attention models like Performer and Lightweight Convolutions show inconsistent scaling behavior, while ALBERT demonstrates negative scaling trends. This finding helps explain why most LLMs maintain relatively standard architectures rather than adopting more exotic variants. Furthermore, Zhai et al. (2022) revealed that ViT reveals that these

| Scaling Law | Primary Goal | Where It Works | Where It Doesn't Work |
|---|---|---|---|
| Kaplan et al. (2020) | Predict model performance scales for model size and dataset size. | Large-scale autoregressive language models (e.g., GPT-style) trained in a pre-asymptotic regime with consistent data quality and full training to convergence. | Breaks down when data is exhausted or models under-train; fails for non-i.i.d. data distributions, transfer learning, and mixture-of-experts models. |
| Hoffmann et al. (2022) | To establish a compute-optimal scaling law by balancing model size and dataset size for minimal loss under fixed compute. | Effective for large dense LMs trained on curated text corpora where both model and data scaling are co-optimized. | Fails under multimodal or non-text regimes, fine-tuning stages, or domains with variable data redundancy; assumes uniform data quality and linear compute cost. |
| Caballero et al. (2023) | To capture multi-regime behavior by allowing scaling exponents to change across parameter or data regimes (piecewise power laws). | Accurately models performance when a clear saturation or "phase change" occurs | Breaks in extrapolation beyond trained regimes; sensitive to breakpoint initialization and noise in small-scale regions. |

Table 4: Comparison of prominent scaling laws under RQ1 in terms of their primary goal, effective regimes, and known failure cases.

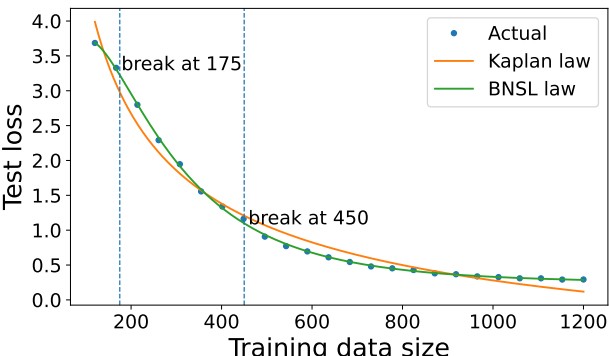

Figure 4: "Double descent" behavior is not well-captured by traditional power-law-based scaling laws such as Kaplan et al. (2020). Piecewise-power-law-based parametric laws (Caballero et al., 2023) (BNSL) captures the smooth transition across different loss regimes.

models exhibit *double saturation*, where performance plateaus at both very low and very high compute levels, suggesting architectural limitations specific to the vision domain (Equation 5). However, as shown by Li et al. (2024a), simply scaling up vision encoders in multimodal models does not consistently improve performance, indicating that architectural scaling benefits are not uniform across modalities.

$$E = a(C + d)^{-b} + c, \tag{5}$$

where $E$ denotes downstream error, $C$ represents compute, and $a, b, c, d$ are fitting parameters.

> **Synthesis and guidelines**
>
> - Architectural bottlenecks vary across domains and compute scales. Transformer inductive biases generalize best under scale.
> - Use architectures with proven scaling profiles (e.g., vanilla Transformer) unless task-specific benefits outweigh risks. For multimodal or domain-specialized setups, consult scaling behavior across compute ranges (Figure 7a).

**RQ3. Data strategies for performance scaling [taxonomy: data scaling]**

Ye et al. (2024) proposed an exponential model for data mixing:

$$L_i(r_{1\ldots M}) = c_i + k_i \exp\left(\sum_{j=1}^{M} t_{ij} r_j\right), \tag{6}$$

while Liu et al. (2024) and Kang et al. (2024) developed proxy models (REGMIX, AUTOSCALE) to pre-optimize mixtures. The Domain-Continual Pretraining (D-CPT) law (Que et al., 2024) provides a theoretical grounding on optimal mixture ratio between general and domain-specific data :

$$L(N, D, r) = E + \frac{A}{N^\alpha} + \frac{B \cdot r^\eta}{D^\beta} + \frac{C}{(r + \epsilon)^\gamma}, \tag{7}$$

where $N$ represents the number of model parameters, $D$ is the dataset size, $r$ is the mixture ratio, $E, A, B, C, \alpha, \beta, \gamma, \eta, \epsilon$ are fitting parameters.

**Where these laws work (and do not).** Table 5 summarizes effective regimes for these scaling laws. The data-mixing formulation is effective for pre-training LLMs on *heterogeneous* corpora where the goal is compute-efficient composition, but becomes cumbersome when the number of data domains is very large due to the growth in fitting parameters. D-CPT (Que et al., 2024) is well-suited for *continual* pre-training, adapting a base model to new domains and supporting long-term updates, yet it is not designed to optimize a single, static mixture from scratch. Figure 5 demonstrates the lack of effectiveness of traditional scaling laws under data-constrained settings, where Chinchilla law fails due to violations of data i.i.d condition in multi-epoch training. As epochs increase, Chinchilla law continues to predict steady loss reductions, whereas the observed test loss saturates; the data-constrained model (in the spirit of Muennighoff et al., 2023) tracks the plateau and recommends allocating budget to *more epochs on smaller models* rather than scaling parameters, capturing the regime dynamics.

> **Synthesis and guidelines**
>
> - Model performance is sensitive to data heterogeneity, mixture ratios, and interaction effects – especially in multi-domain or continual settings.
> - Use mixture-aware laws (e.g., Ye et al. (2024)) for heterogeneous pre-training; prefer D-CPT for continual updates; and switch to data-constrained prescriptions when repeated exposures drive saturation (as in Figure 5).
> - Replace manual corpus aggregation with predictive data mixing. Use D-CPT law when adapting to specific domains. Figure 7a outlines strategy paths based on data availability and domain constraints.

**RQ4. Test-time scaling for better scaling efficiency [taxonomy: model scaling $\rightarrow$ inference scaling]**

Recent research examining the relationship between test-time computation and model size scaling has revealed key insights. Brown et al. (2024) proposed that repeated sampling during inference significantly enhances model performance, with coverage $C$ (fraction of problems solved) following an exponentiated power law relationship with the number of samples $k$, $\log(C) = ak^{-b}$, where $a, b$ are fitting parameters. Further exploration by Wu et al. (2024) suggested that employing sophisticated test-time computation strategies (such as iterative refinement or tree search) with smaller models may be more cost-effective than using larger models with simple inference methods. Their work establishes a relationship between inference computational budget and optimal model size for compute-efficient inference, expressed as $\log_{10}(C) = 1.19 \log_{10}(N) + 2.03$.

| Scaling Law | Primary Goal | Where It Works | Where It Doesn't Work |
|---|---|---|---|
| Ye et al. (2024) | Predict optimal data compositions for compute-efficient pre-training. | Pre-training LLMs, especially for optimizing heterogeneous datasets. | Scenarios with a very large number of data domains, due to the complexity and number of fitting parameters. |
| Que et al. (2024) | Balance general and domain-specific data for continual pre-training. | Adapting pre-trained models to new domains efficiently and for long-term updates. | Optimizing a single, static data mixture from scratch, as it is specifically designed for continual pre-training. |

Table 5: Comparison of prominent scaling laws under RQ3 in terms of their primary goal, effective regimes, and known failure cases.

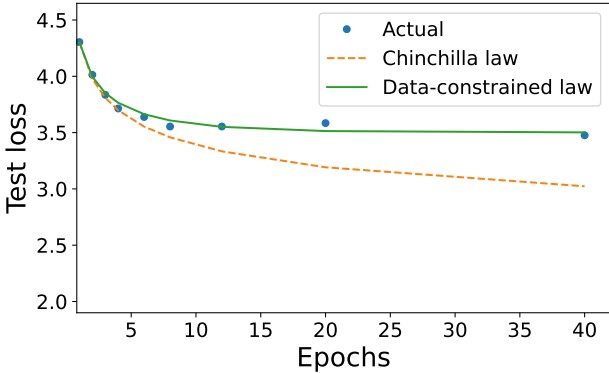

Figure 5: Importance of data-constrained scaling laws, where traditional scaling laws fail to incorporate the data-dependence factors in multi-epoch pre-training.

---

**Synthesis and guidelines**

- Inference scaling offers a complementary path to performance, particularly where model reuse is desired but compute cost must remain low.
- Use adaptive compute, retrieval augmentation, or tree search for high-value queries. Integrate test-time scaling laws into deployment workflows (Figure 7b).

---

**RQ5. Scaling behaviors of model fine-tuning [taxonomy: model scaling → post-training scaling]**

Fine-tuning scaling reflects how pre-trained models adapt across tasks and domains. Hernandez et al. (2021) introduced a transfer scaling law based on effective data transferred $D_t$:

$$D_t(D_f, N) = k(D_f)^\alpha (N)^\beta, \tag{8}$$

while Lin et al. (2024b) refined this with a rectified law:

$$L(D) = \frac{B}{D_t + D^\beta} + E, \tag{9}$$

modeling diminishing returns from fine-tuning beyond a pre-learned threshold. In vision, Abnar et al. (2021) linked downstream error to upstream error:

$$e_{DS} = k(e_{US})^a + c, \tag{10}$$

and Mikami et al. (2021) connected downstream accuracy to synthetic pretraining data size:

$$e_{DS} = aD^{-\alpha} + c. \tag{11}$$

*FLOPS to Loss to Performance* (FLP) method (Chen et al., 2024c) predicted downstream performance from pretraining FLOPs, and Zhang et al. (2024) showed LoRA scales nonlinearly under PEFT:

$$\hat{L}(X, D_f) = A \times \frac{1}{X^\alpha} \times \frac{1}{D_f^\beta} + E. \tag{12}$$

> **Synthesis and guidelines**
>
> - Transferability scales with both model size and pretraining loss, but task difficulty, data availability, and adaptation type mediate returns.
> - Use FLP or rectified laws to estimate post-training gains. Prefer PEFT for low-resource settings; switch to full fine-tuning when compute permits. For domain adaptation, apply D-CPT strategies (Figure 7a).

**RQ6. Scaling efficiency and performance for sparse and efficient models [taxonomy: model scaling $\rightarrow$ model compression]**

As the demand for resource-efficient models grows, sparse architectures such as pruned networks and MoEs have emerged as promising alternatives to dense Transformers. These models aim to preserve the performance benefits of scale while reducing compute and memory overhead. Frantar et al. (2023) proposed a general sparse scaling law showing that sparsity acts as a multiplicative efficiency factor rather than changing the fundamental scaling behavior:

$$L(S, N, D) = (a_S(1 - S)^{b_S} + c_S) \cdot \left(\frac{1}{N}\right)^{b_N} + \left(\frac{a_D}{D}\right)^{b_D} + c, \tag{13}$$

where $S$ is sparsity, $N$ is the number of non-zero parameters, and $D$ is dataset size. In MoE models, where only a subset of parameters is activated per input, Clark et al. (2022) proposed a loss scaling relationship incorporating both model size and expert count:

$$\log L = a \log N + b \log E + c \log N \cdot \log E + d, \tag{14}$$

with $E$ denoting the expansion factor. This formulation was extended by Yun et al. (2024) to include dataset size:

$$\log L(N, D, E) = \log \left(\frac{a}{N^\alpha} + \frac{b}{E^\beta} + \frac{c}{D^\gamma} + f\right) \\ + d \log N \log E \tag{15}$$

These results emphasize that scaling MoEs effectively requires balancing expert granularity with sufficient training data. Toward this, Krajewski et al. (2024) introduced a granularity parameter $G$ to refine the Chinchilla-style formulation:

$$\mathcal{L}(N, D, G) = c + \left(\frac{g}{G^\gamma} + a\right) \frac{1}{N^\alpha} + \frac{b}{D^\beta}. \tag{16}$$

In parallel, structured pruning approaches have been formalized through the $P^2$ law (Chen et al., 2024b), which relates post-pruning loss to pre-pruning model size $N_0$, pruning ratio $\rho$, and post-training token count $D$:

$$L(N_0, D, \rho, L_0) = L_0 + \left(\frac{1}{\rho}\right)^\gamma \left(\frac{1}{N_0}\right)^\delta \left(\frac{N_C}{N_0^\alpha} + \frac{D_C}{D^\beta} + E\right), \tag{17}$$

where $L_0$ is the uncompressed model loss, $\rho$ is the pruning rate, $N_0$ is the pre-pruning model size, $D$ represents the number of post-training tokens, and $N_C, D_C, E, \alpha, \beta, \gamma$ are fitting parameters.

**How the sparse/MoE scaling functions evolved.** Table 6 traces a progression from (i) *sparsity-as-efficiency* laws for pruned *dense* networks (Frantar et al., 2023), where $S$ multiplies standard size/data terms, to (ii) *MoE interaction* laws that explicitly model the synergy between total parameters $N$ and expert count $E$ (including $N \times E$ interaction terms) (Yun et al., 2024), and finally to (iii) *granularity-aware* MoE

| Scaling Law | Primary Goal | Where It Works | Where It Doesn't Work |
|---|---|---|---|
| Frantar et al. (2023) | To model sparsity (S) as a multiplicative efficiency factor on top of standard scaling laws. | For models with static, unstructured sparsity (e.g., pruned networks) to find the optimal sparsity for a given compute budget. | For dynamic, structured sparsity like Mixture-of-Experts (MoE), as it doesn't capture the complex interaction between experts. |
| Clark et al. (2022); Yun et al. (2024) | To predict MoE performance by modeling the synergistic interaction between model size (N) and the number of experts (E). | Specifically for MoE architectures to balance total parameters against the number of experts for optimal performance. | For standard dense models or pruned networks where the concept of "experts" does not apply. |
| Krajewski et al. (2024) | To refine MoE scaling by introducing a granularity parameter (G), accounting for expert size and specialization. | For advanced MoE optimization where expert size is tunable, helping decide between many small experts vs. fewer large ones. | When a quantifiable "granularity" metric is unavailable, making it too specific for general MoE models. |

Table 6: Comparison of prominent scaling laws under RQ6 in terms of their primary goal, effective regimes, and known failure cases.

laws (Krajewski et al., 2024) that introduce a controllable $G$ capturing expert specialization and yielding a compute-optimal envelope across granularities.

**Where these laws work (and do not).** From Table 6: the Frantar et al. (2023) formulation works well for models with *static, unstructured* sparsity (e.g., pruned nets) to select optimal sparsity at a given budget, but it does *not* capture the complex interaction patterns of MoE routing. The Clark et al. (2022); Yun et al. (2024) MoE laws are effective *within* MoE architectures to balance $N$ against the number of experts $E$, yet they are not applicable to standard dense/pruned models without experts. The granularity-based law of Krajewski et al. (2024) is most useful for *advanced MoE optimization* when expert size/specialization is tunable, but it becomes less general when a measurable $G$ is unavailable. Figure 6 shows training-budget–vs–loss curves where the *dense* Transformer (dashed) underperforms a family of MoE models with varying $G$ at the *same* FLOPs, and the *optimal MoE envelope* (solid) lies strictly below the dense curve across budgets. This exemplifies a case where dense scaling laws fail to capture attainable efficiency, while the granularity-aware MoE law correctly predicts that increasing expert granularity (up to a budget-dependent optimum) yields better compute–quality trade-offs.

---

**Synthesis and guidelines**

- Sparse models are scaling-compliant but require careful routing (MoE) and token-budget tuning (pruning) to outperform dense counterparts.
- Match regime to law (Table 6): choose sparsity-efficiency laws for static pruning; interaction laws for MoE size–expert balancing; and granularity-aware laws when expert specialization is a tunable knob and compute-optimal envelopes matter.
- Use MoEs for general-purpose LLMs under compute limits. Apply pruning for deployment constraints. For efficient inference, refer to Figure 7b.

---

**RQ7. Model scaling with low-precision quantization [taxonomy: model scaling → model compression → quantization]**

According to Dettmers & Zettlemoyer (2023), 4-bit precision appears to be the optimal sweet spot for maximizing model performance while minimizing model size. Additionally, research on scaling with mixed quantization (Cao et al., 2024), demonstrated that larger models can handle higher quantization ratios while maintaining performance, following an exponential relationship where larger models require exponentially fewer high-precision components to maintain a given performance level. Kumar et al. (2024) developed a unified scaling law (Equation 18) that predicts both training and post-training quantization effects. It further

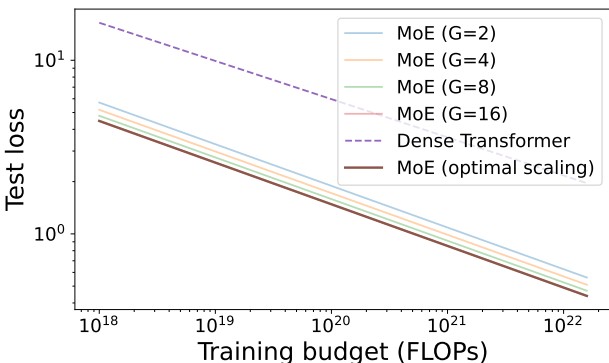

Figure 6: Compute optimal scaling of mixture-of-experts are often better than dense transformers – highlighting the importance of sparser activations for achieving better performance at same compute cost. Scaling shown in log-log scale.

suggests that effects of quantizing weights, activations, and attention during training are independent and multiplicative.

$$L(N, D, P_w, P_a, P_{\text{kv}}, P_{\text{post}}) = AN_{\text{eff}}^{-\alpha} + BD^{-\beta} + E + \delta_{PTQ}, \tag{18}$$

where $P_w, P_a, P_{kv}$ denote training precision of weights, activations and attentions, respectively, $P_{post}$ denote end-time weight-precision, $\delta_{PTQ}$ denotes loss due to post training quantization, and $\alpha, \beta$ are fitting parameters.

> **Synthesis and guidelines**
>
> - Scaling-aware quantization reduces memory while preserving performance. Larger models generalize better to low precision.
> - Apply mixed-precision for inference. Use quantization-aware training for smaller models. Refer to post-training strategies (Figure 7b) to guide compression.

**RQ8. Beyond modalities: scaling for multimodal models [taxonomy: model scaling → multimodal models]**

Multimodal scaling behavior builds upon, but does not replicate, unimodal trends. Henighan et al. (2020) first proposed multimodal scaling using $L(x) = Ax^{-\alpha} + B$, where $x$ represents model size, data, or compute. Alabdulmohsin et al. (2022) refined this into a more flexible sigmoid-like form:

$$\frac{L_x - L_\infty}{(L_0 - L_x)^\alpha} = \beta x^c, \tag{19}$$

allowing transitions across saturation regimes. Aghajanyan et al. (2023) observed that smaller multimodal models exhibit competition between modalities, while larger models cross a "competition barrier" and become synergistic. They proposed a bimodal generalization of the Chinchilla law:

$$\begin{aligned}
\mathcal{L}(N, D_i, D_j) = &\left[ \frac{\mathcal{L}(N, D_i) + \mathcal{L}(N, D_j)}{2} \right] \\
&- C_{i,j} + \frac{A_{i,j}}{N^{\alpha_{i,j}}} + \frac{B_{i,j}}{|D_i| + |D_j|^{\beta_{i,j}}},
\end{aligned} \tag{20}$$

where $C_{i,j}$ captures the degree of positive interaction between modalities $i$ and $j$.

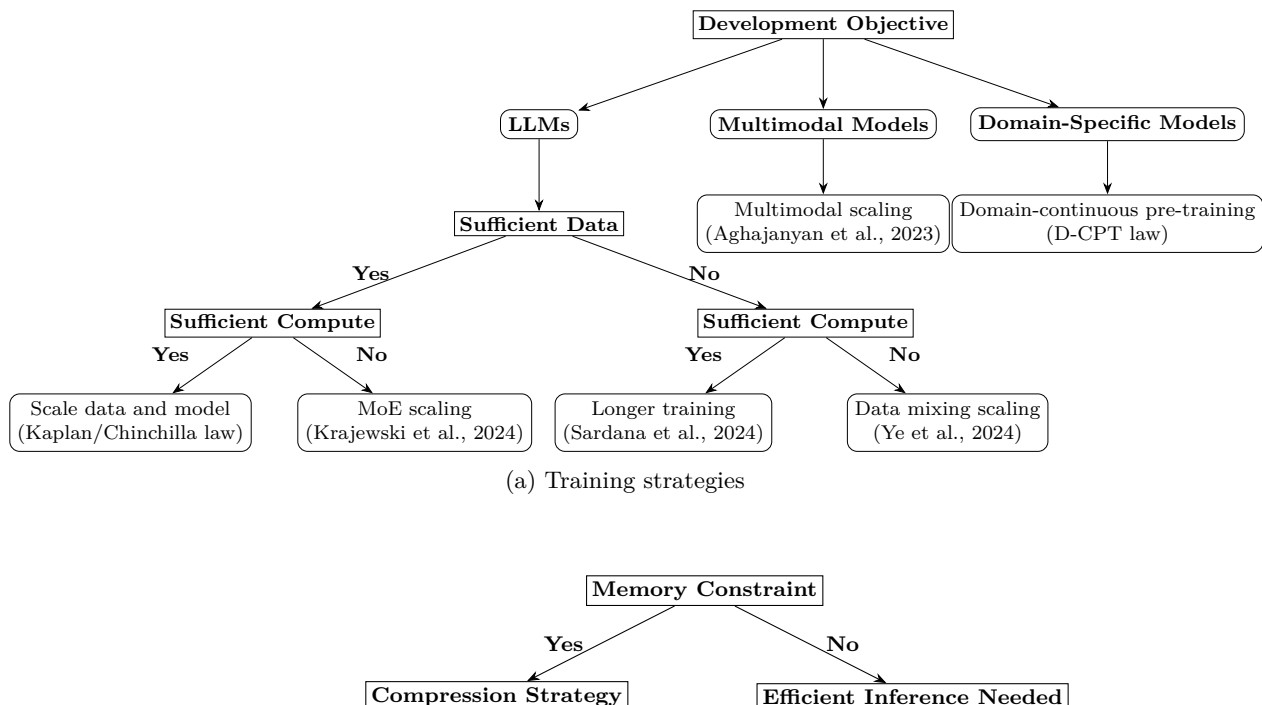

(a) Training strategies

(b) Inference strategies

Figure 7: Practical roadmap summarizing training and inference strategies grounded in our eight research questions and taxonomy branches. (a) Training scaling strategies can be utilized for pre-training or fine-tuning unimodal and multimodal foundational and domain-adapted models. (b) Post-training inference strategies can be followed to ensure that the model is utilized efficiently for the downstream applications.

> **Synthesis and guidelines**
>
> - Multimodal scaling is governed by modality alignment and architectural balance more than raw model size.
> - Ensure models are sufficiently large to benefit from synergy across modalities. Prioritize modality balance in architecture and high-quality aligned datasets over isolated scaling. Refer to Figure 7a when designing multimodal pretraining pipelines.

> **Cross-RQ synthesis**
>
> - Data-efficient scaling (RQ1, RQ3, RQ5) consistently beats brute-force model expansion, as shown in Hu et al. (2024); Sardana et al. (2024).
> - Architectural innovations (RQ2, RQ6) tend to scale poorly unless paired with precise training heuristics (e.g., expert routing in MoEs).
> - Inference-aware scaling (RQ4, RQ7) enables small models to rival larger ones but is rarely included in current scaling laws - a key research gap.

While the research questions synthesized above highlight the strengths and practical applications of neural scaling laws, they also expose several limitations, especially in their generalizability, reliability under constraints and applicability to modern model designs. In the next section, we critically examine these limitations and discuss the foundational assumptions that may no longer hold as models evolve.

# 5 Criticisms of scaling laws

Diaz & Madaio (2024) challenged the generalizability of neural scaling laws, arguing that they fail in diverse real-world AI applications. They argued that scaling laws do not always hold when AI models serve heterogeneous populations with conflicting criteria for model performance. Larger datasets inherently reflect diverse communities, making it difficult to optimize a single model for all users. Similar to issues in multilingual AI, increasing data diversity often leads to performance degradation rather than improvement. Universal evaluation metrics are inadequate for capturing these complexities, potentially reinforcing biases against underrepresented groups. The authors further argued that smaller, localized AI models may be more effective for specific communities, highlighting the need to move beyond one-size-fits-all scaling assumptions.

Beyond dataset expansion, data pruning contradicts traditional scaling laws by demonstrating that performance improvements do not always require exponentially more data. Strategic pruning achieves comparable or superior results with significantly fewer training samples (Sorscher et al., 2023). Not all data contributes equally, and selecting the most informative examples enables more efficient learning. Experimental validation on CIFAR-10, SVHN, and ImageNet shows that careful dataset curation can surpass traditional power-law improvements, questioning the necessity of brute-force scaling.

Despite their significant impact, many studies on scaling laws suffer from limited reproducibility (see Table 15 in Appendix B) due to proprietary datasets, undisclosed hyperparameters, and undocumented training methodologies. The inability to replicate results across different computing environments raises concerns about their robustness. Large-scale experiments conducted by industry labs often depend on private infrastructure, making independent verification challenging. This lack of transparency undermines the reliability of scaling law claims and highlights the urgent need for open benchmarks and standardized evaluation frameworks to ensure reproducibility. Furthermore, the field's avoidance of rigorous scaling exponent analysis constitutes a critical oversight. While exponents indeed vary across models, datasets, and hyperparameters, this variability demands investigation rather than dismissal. This deliberate analytical gap undermines confidence in extrapolation claims and raises questions about whether observed scaling behaviors represent genuine properties or experimental artifacts.

Goel et al. (2025) further questioned the validity and practicality of neural scaling laws by emphasizing their computational inefficiency, environmental unsustainability, and deployment constraints. They argued that scaling approaches often overlook diminishing returns, particularly the disproportionate computational and carbon emission costs associated with incremental performance gains. For example, their analysis (illustrated in Figure 8) showed that modest improvements in model performance demand exponential increases in resource usage, thus challenging the assumption that scaling laws represent sustainable progress. To address these limitations, they proposed a systematic shift towards "downscaling", advocating for smaller, domain-adapted models trained with carefully curated datasets. This downscaling approach not only mitigates environmental impacts but also promotes wider accessibility by lowering computational and financial barriers. Their study highlights an urgent need for revising the prevailing scaling paradigm to ensure more efficient and responsible AI development.

# 6 Beyond Scale: Future Directions for Practical and Sustainable AI

While neural scaling laws have provided valuable insights, they often *mis-predict* in settings that deviate from the classic, dense, pre-training-on-i.i.d.-text paradigm. In particular, real deployments involve multiple competing constraints (latency, energy, memory), non-i.i.d. data (duplicates, domain drift), modular inference (retrieval, routing, MoE), and objective functions beyond cross-entropy. To make scaling laws decision-useful (we highlight a guided roadmap toward better scaling practices in Figure 9), we must move from single-metric power-laws to *modular, cost-aware, and inference-aware* formulations that generalize across data quality, domains, and hardware.

**Reframing scaling laws for real-world constraints.** Future scaling laws must account for compute budgets, hardware latency, and energy consumption. This includes integrating training–inference trade-offs, evaluating real-world performance under quantization or pruning, and predicting effectiveness across

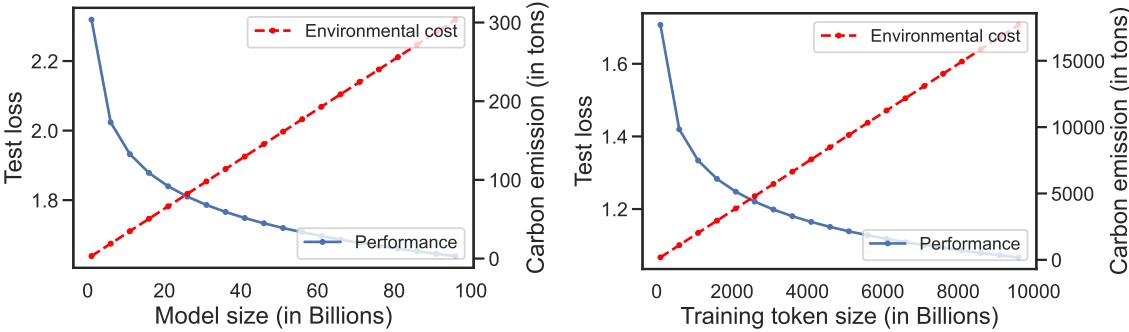

(a) Environment costs of increasing model size     (b) Environment costs of increasing training data size

Figure 8: As synthesized by Goel et al. (2025), performance scales logarithmically with model and pre-training data sizes, whereas, environmental factors like Carbon emission during pre-training increases linearly. Outgrowing environmental cost highlights the infeasibility of power-law governing scaling laws for scaling up beyond a fixed limit.

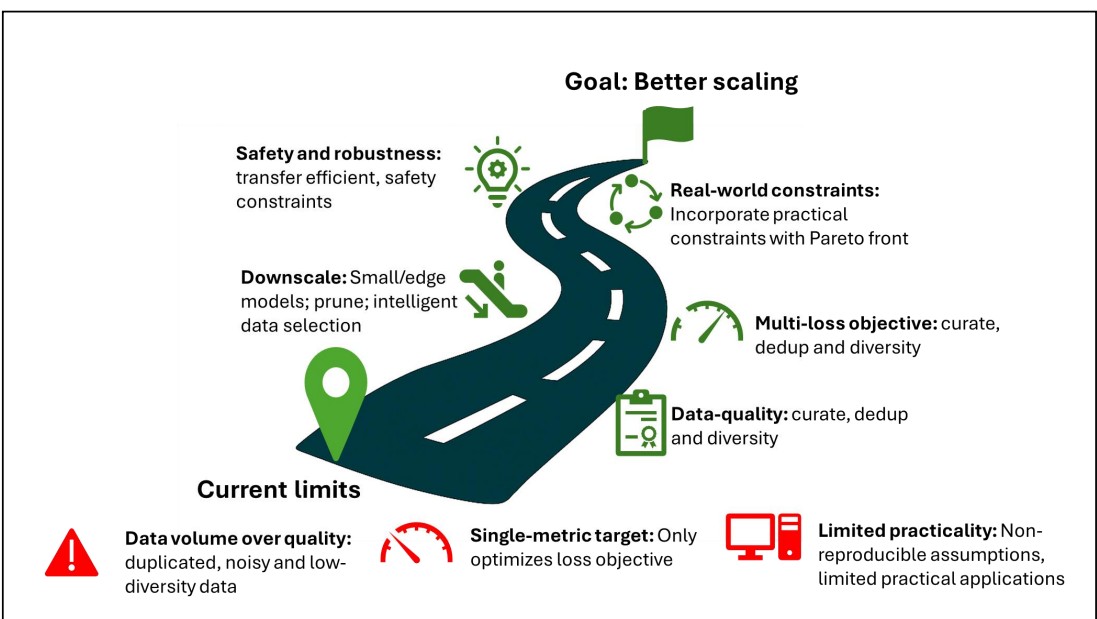

Figure 9: **Roadmap from "what exists" to "better scaling."** The diagram contrasts *current limits* (left) – data volume over quality, training-only laws, single-metric targets, and dense-only assumptions, with a *road to better scaling* (right) that prioritizes: (1) **Data Quality** (curate diverse high-quality datasets, no synthetic duplicated data); (2) **Downscaling** for small/edge models with quantization and pruning; (3) Incorporating **Compute & Energy** constraints (budget, carbon, memory) while deriving the optimal configuration; (4) **Multi-objective** optimization (Pareto trade-offs over loss/latency/energy/robustness); (5) **Robustness & Safety** (reliability of scaling laws in practical applications).

resource-constrained environments. Concretely, we advocate reporting *Pareto fronts* over (*loss*, *latency*, *energy*, *memory*) and learning *cost-aware exponents* that vary with hardware and batch/sequence shapes. Empirically, the same architecture can move across Pareto fronts as tokenization, parallelism strategy, or KV-cache policy changes; future laws should *parameterize* these deployment knobs.

**Designing for downscaling.** Rather than building ever-larger models, the field should invest in scaling laws for *small* language models trained with optimal data, sparsity, and inference strategies. The emergence

of 1–3B parameter models that rival 13B+ models (Hu et al., 2024) highlights the viability of compact yet performant systems. Building on this, Goel et al. (2025) formalize a *downscaling law* that predicts when an *ensemble of smaller, pruned models* will outperform a single larger model under the *same* compute budget. Let $L_0$ denote the loss of a base model and $n$ the number of models in a deep ensemble; the expected ensemble loss scales as $L(n) = L_0 - b + \frac{b}{n^a}$, with positive, task/model-specific fit parameters $a, b$. Combining this with the $P^2$ post-pruning law (Chen et al., 2024b) (capturing the effect of pruning ratio and post-training tokens on loss), they derive a compute-parity condition ensuring that an $n$-way ensemble of pruned models attains strictly lower loss than the base model at equal cost: $\left(\frac{n^a-1}{n^{a+\gamma}}\right)(n-1)^\gamma \geq \frac{1}{b\,N_0^\delta}$, where $N_0$ is the pre-pruning parameter count and $(\gamma, \delta)$ are the pruning/finetuning exponents inherited from $P^2$. Practically, this law trades *model size*, *pruning/finetuning*, and *ensemble width* to achieve the best attainable quality at edge/mobile budgets. These downscaling laws complement our emphasis on *token quality*, routing capacity, and quantization depth by offering a closed-form criterion for selecting the number of pruned/quantized replicas that optimizes quality under tight deployment constraints.

**Multi-objective scaling optimization.**  Current scaling laws often predict accuracy at scale but ignore trade-offs between accuracy, compute, and robustness. Future work should develop *multi-objective scaling frameworks* that balance these factors to guide architecture and dataset design more holistically. We recommend framing scaling as a *multi-objective program* with constraints on worst-case error/robustness, privacy, and environmental factors. Analytically, this suggests replacing single power-laws with *families* of conditional laws $L(N, D, \mathcal{C})$ indexed by constraint bundle $\mathcal{C}$, and optimizing over Pareto-efficient frontiers. For instance, we can propose an augmenting loss function with compute cost, defining a multi-objective target $\min(L + \lambda \mathcal{C})$ where $\mathcal{C}$ is compute or energy expenditure. This shifts scaling evaluation toward efficient, sustainable regimes.

**Inference-aware and modular scaling laws.**  Traditional scaling laws assume fixed inference procedures. However, our synthesis in **RQ4** and **RQ7** shows that test-time compute allocation via sampling, retrieval, or routing can drastically affect performance. Future scaling formulations should modularize inference and allow flexible compute allocation per task or query. We propose *inference-aware scaling* that introduces explicit knobs for retrieval volume, routing granularity (MoE experts per token), and speculative/iterative decoding depth. The resulting laws predict quality as a joint function of training tokens, model parameters, *and* inference budget. This also enables *per-query* optimal policies (e.g., adaptively increase retrieval or expert budget for hard inputs).

**Data quality over quantity.**  Instead of expanding datasets indiscriminately, laws like REGMIX (Liu et al., 2024) and D-CPT (Que et al., 2024) emphasize optimized data composition. Future models should prioritize informative examples and track dataset efficiency across tasks. We argue for *quality-weighted token accounting*: let $D_{\text{eff}} = \sum_i w_i$ with weights $w_i$ reflecting information value (deduplication, diversity, noise filtering, curriculum). Scaling exponents should be learned over $D_{\text{eff}}$ instead of raw token counts. This unifies observations that carefully curated or repeated data can outperform naive laws and explains regime changes (e.g., when data pruning "beats" the baseline power-law).

**Domain-specific laws (RL, diffusion, robotics).**  Many domains break classic power-law assumptions (more descriptions in Appendix A) In *RL*, non-stationarity, horizon length, and exploration alter sample complexity; scaling should condition on horizon and environment entropy. In *diffusion* models, quality depends on training scale *and* sampling steps/schedulers; laws must tie generation compute to FID/IS and downstream metrics. In *robotics*, data is interactive, multi-modal, and safety-bound; scaling should model demonstration quality, sim2real gaps, and control latency. Dedicated, measurement-grounded laws will avoid the "one-size-fits-all" fallacy.

**Transfer, safety, and robustness at scale.**  Power-laws fitted on pre-training losses can overestimate downstream reliability. Future laws should co-model *transfer efficiency*, interference between tasks (negative transfer), and *safety constraints* (adversarial robustness, specification gaming). This suggests hierarchical scaling: pre-training curves that map into downstream task families via transfer operators, with robustness penalties that widen at larger scales unless mitigated by data curation or alignment.

**What "good" future laws look like.** A practical scaling law should: (i) admit *deployment knobs* (latency, memory, precision, cache policy), (ii) be *inference-aware* (retrieval/routing budgets), (iii) operate on *quality-weighted tokens*, and (iv) expose *Pareto fronts* and constraint-aware optima. Such laws turn scaling from curve-fitting into *design guidance* for real systems, across both upscaling and downscaling regimes.

## 7 Conclusion

This survey provided a comprehensive analysis of neural scaling laws, exploring their theoretical foundations, empirical findings, and practical implications. It synthesized insights across various modalities, including language, vision, multimodal, and reinforcement learning, to uncover common trends and deviations from traditional power-law scaling. While early research established predictable relationships between model size, dataset volume, and computational resources, more recent studies have shown that these relationships are not universally applicable. Sparse architectures, retrieval-augmented models, and domain-specific adaptations often exhibit distinct scaling behaviors, challenging the notion of uniform scalability. Furthermore, advancements in fine-tuning, data pruning, and efficient inference strategies have introduced new perspectives on compute-optimal scaling. Despite their significance, scaling laws remain an evolving area of research, requiring further refinement to address real-world deployment challenges and architectural innovations.

## Limitations

While this survey provides a broad synthesis of neural scaling laws, it primarily focuses on model size, data scaling, and compute efficiency. Other important aspects, such as hardware constraints, energy consumption, and the environmental impact of large-scale AI training, are not deeply explored. Another limitation is the reliance on prior empirical findings, which may introduce variability due to differing experimental setups and proprietary datasets. Without access to fully reproducible scaling law experiments, some conclusions remain dependent on the methodologies employed in original studies.

## Acknowledgments

T. Chakraborty acknowledges the support of the IBM-IITD AI Horizons network and Rajiv Khemani Young Faculty Chair Professorship in Artificial Intelligence. He acknowledges the support of Google GCP Grant for providing the necessary computational resources.

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

# A Detailed scaling laws

## A.1 Scaling laws of language models

Kaplan et al. (2020) suggested that larger LMs improve performance by reducing loss through power-law scaling. However, this view evolved when studies showed that many large models were undertrained, and data

scaling plays an equally crucial role in compute efficiency (Hoffmann et al., 2022). More recent breakthroughs challenged traditional scaling assumptions. Broken Neural Scaling Law (BNSL) introduced non-monotonic trends, meaning that model performance can sometimes worsen before improving, depending on dataset thresholds and architectural bottlenecks (Caballero et al., 2023). Another exciting development came from small LMs, where optimized training strategies, such as a higher data-to-parameter ratio and adaptive learning schedules, enable models ranging from 1.2B to 2.4B parameters to rival significantly larger 7B-13B models (Hu et al., 2024). These findings reshape the fundamental assumptions of scaling laws, proving that strategic training can outperform brute-force model expansion.

| Modality | Paper | Key insights | Applicability |
|---|---|---|---|
| Language | Kaplan et al. (2020) | Larger models are more sample-efficient, needing fewer training examples to generalize well. | Predicts model loss decreases with increasing parameters, used in early LMs like GPT-3. |
| | Hoffmann et al. (2022) | The best performance comes from balancing model size and data, rather than just increasing parameters. | Balances compute, model size, and dataset size for optimal efficiency, as seen in Chinchilla. |
| | Caballero et al. (2023) | Performance does not always improve smoothly; there are inflection points where scaling stops working. | Identifies phase transitions, minimum data thresholds, and unpredictability in scaling behavior. |
| | Hu et al. (2024) | Smaller models with better training can rival much larger models. | Demonstrates that smaller models with optimized training can outperform larger undertrained models. |
| Vision | Zhai et al. (2022) | ViTs follow power-law scaling but plateau at extreme compute levels, with benefits primarily seen in datasets >1B images. | Image classification, object detection, large-scale vision datasets. |
| Multimodal | Aghajanyan et al. (2023) | Multimodal models experience competition at smaller scales but transition into synergy as model and token count grow. | Multimodal learning, mixed-modal generative models, cross-domain AI. |
| | Li et al. (2024a) | Scaling vision encoders in vision-language models does not always improve performance, reinforcing the importance of data quality over raw scaling. | Vision-language models, image-text alignment, multimodal scaling challenges. |

Table 7: Critical neural scaling laws for language, vision and multimodal models.

## A.2 Scaling laws in other modalities

In computer vision, ViTs exhibit power-law scaling when model size, compute, and data grow together, but their performance plateaus at extreme compute levels, with noticeable gains only when trained on datasets exceeding 1B images (Zhai et al., 2022). Meanwhile, studies on scaling law extrapolation revealed that while larger models generally scale better, their efficiency declines at extreme sizes, requiring new training strategies to maintain performance (Alabdulmohsin et al., 2022). In multimodal learning, an interesting phenomenon called the "competition barrier" has been observed where at smaller scales different input modalities compete for model capacity, but as models grow, they shift into a synergistic state, enabling accurate performance predictions based on model size and token count (Aghajanyan et al., 2023).

However, not all scaling trends align with expectations. Contrary to the assumption that larger is always better, scaling vision encoders in vision-language models can sometimes degrade performance, highlighting the fact that data quality and modality alignment are more critical than brute-force scaling (Li et al., 2024a). These findings collectively emphasize that scaling laws are domain-dependent – optimal scaling strategies require a careful balance between compute efficiency, dataset quality, and architecture rather than simply

| Paper | Key insights | Applicability |
|---|---|---|
| Zhai et al. (2022) | ViTs follow power-law scaling but plateau at extreme compute levels, with benefits primarily seen in datasets >1B images. | Image classification, object detection, large-scale vision datasets. |
| Aghajanyan et al. (2023) | Multimodal models experience competition at smaller scales but transition into synergy as model and token count grow, following a "competition barrier." | Multimodal learning, mixed-modal generative models, cross-domain AI. |
| Li et al. (2024a) | Scaling vision encoders in vision-language models (VLMs) does not always improve performance, reinforcing the importance of data quality over raw scaling. | Vision-language models, image-text alignment, multimodal scaling challenges. |

Table 8: Summary of key insights found in scaling laws paper for computer vision and multimodal domains.

increasing model size. Table 7 summarizes the scaling laws of pre-trained models for language and other modalities.

## A.3 Scaling laws for domain adaptation

Pre-training and fine-tuning techniques have accelerated the adoption of large-scale neural models, yet the extent to which these models transfer across tasks and domains remains a key research question tied to scaling principles. Studies show that transfer learning follows a power-law where pre-training amplifies fine-tuning effectiveness, especially in small data regimes. Even with limited downstream data, larger models benefit significantly from pre-training, improving generalization (Hernandez et al., 2021). In vision, pre-training saturation occurs due to upstream-downstream interactions, rather than just task complexity. Lower network layers quickly specialize in simple tasks, while higher layers adapt to complex downstream objectives (Abnar et al., 2021). Similarly, in synthetic-to-real transfer, larger models consistently reduce transfer gaps, enhancing generalization across domains (Mikami et al., 2021).

Fine-tuning strategies scale differently depending on dataset size. Parameter-efficient fine-tuning (PEFT) techniques like low-rank adaptation (LoRA) (Hu et al., 2021) and Prompt-tuning, both are well-suited for small datasets, but LoRA performs best for mid-sized datasets, and full fine-tuning is most effective for large datasets. However, PEFT methods provide better generalization in large models, making them attractive alternatives to full-scale fine-tuning (Zhang et al., 2024).

Scaling laws are also being utilized to accurately predict the fine-tuning performance of models. The FLP method (Chen et al., 2024c) estimates pre-training loss from FLOPs, enabling accurate forecasts of downstream performance, particularly in models up to 13B parameters. Further refinements like FLP-M improve mixed-dataset predictions and better capture emergent abilities in large models. Finally, the Rectified scaling law (Lin et al., 2024b) introduces a two-phase fine-tuning transition, where early-stage adaptation is slow before shifting into a power-law improvement phase. This discovery enables compute-efficient model selection using the "Accept then Stop" (AtS) algorithm to terminate training at optimal points.

We summarize these findings in Table 9, suggesting that transfer learning is highly scalable, but effective scaling requires precise tuning strategies rather than just increasing model size.

## A.4 Scaling laws for model inference

Simply scaling up models is not always the best way to improve model performance. Chen et al. (2024a) suggested that more efficient test-time compute strategies can dramatically reduce inference costs while maintaining or even exceeding performance. Instead of blindly increasing LLM calls, they further suggested

| Paper | Key insights | Applicability |
|---|---|---|
| Hernandez et al. (2021) | Pre-training amplifies fine-tuning, particularly for small datasets, and benefits larger models even under data constraints. | Transfer learning, pre-training optimization, few-shot learning. |
| Abnar et al. (2021) | Large-scale pre-training improves downstream performance, but effectiveness depends on upstream-downstream interactions, not task complexity. | Vision transfer learning, upstream-downstream performance interactions. |
| Zhang et al. (2024) | Optimal fine-tuning strategy depends on dataset size: PEFT for small, LoRA for mid-scale, and full fine-tuning for large-scale datasets. | Fine-tuning strategies, parameter-efficient tuning, LoRA, full fine-tuning. |
| Lin et al. (2024b) | Fine-tuning follows a two-phase transition: slow early adaptation followed by power-law improvements, guiding compute-efficient model selection. | Compute-efficient fine-tuning, early stopping, model selection strategies. |

Table 9: Key highlights from scaling of fine-tuned and domain-adapted models.

for allocating resources based on query complexity, ensuring that harder queries receive more compute while simpler ones use fewer resources. The importance of test-time compute strategies becomes even clearer when dealing with complex reasoning tasks. While sequential modifications work well for simple queries, parallel sampling and tree search dramatically improve results on harder tasks. Adaptive compute-optimal techniques have been shown to reduce computational costs by $4\times$ without degrading performance, allowing smaller models with optimized inference strategies to surpass much larger models (Snell et al., 2024; Brown et al., 2024). Advanced inference approaches, such as REBASE tree search (Wu et al., 2024), further push the boundaries of efficiency, enabling small models to perform on par with significantly larger ones.

Another breakthrough came from retrieval augmented models, where increasing the datastore size consistently improves performance without hitting saturation (Shao et al., 2024). This allows smaller models to outperform much larger ones on knowledge-intensive tasks, reinforcing that external datastores provide a more efficient alternative to memorizing information in model parameters.

## A.5 Scaling laws for efficient models

Scaling laws have expanded beyond simple parameter growth, introducing new methods to optimize routing, sparsity, pruning, and quantization for efficient LLM scaling. Routing-based models benefit from optimized expert selection, but their returns diminish at extreme scales, requiring careful expert configuration (Clark et al., 2022). In contrast, fine-grained MoE models consistently outperform dense transformers, achieving up to $40\times$ compute efficiency gains when expert granularity is properly tuned (Krajewski et al., 2024). However, balancing the number of experts ($E$) is crucial, where models with 4-8 experts offer superior inference efficiency, but require $2.5-3.5\times$ more training resources, making 16-32 expert models more practical when combined with extensive training data (Yun et al., 2024). Sparse model scaling offers another efficiency boost. Research has demonstrated that higher sparsity enables effective model scaling, allowing $2.15\times$ more parameters at 75% sparsity, improving training efficiency while maintaining performance (Frantar et al., 2023). Additionally, pruning laws ($P^2$ scaling laws) predict that excessive post-training data does not always improve performance, helping optimize resource allocation in pruned models (Chen et al., 2024b). Dettmers & Zettlemoyer (2023) showed that 4-bit quantization provides the best trade-off between accuracy and model size, optimizing zero-shot performance while reducing storage costs. Larger models tolerate lower precision better, following an exponential scaling law where fewer high-precision components are needed to retain performance (Cao et al., 2024). Meanwhile, training precision scales logarithmically with compute budgets, with 7-8 bits being optimal for balancing size, accuracy, and efficiency (Kumar et al., 2024). Recent reserach has expanded into distillation as well, developing a mathematical framework that predicts how well

| Paper | Key insights | Applicability |
|---|---|---|
| Brown et al. (2024) | Adaptive test-time compute strategies reduce computational costs by 4× while maintaining performance, enabling smaller models to compete with much larger ones. | Test-time compute efficiency, inference cost reduction, compute-limited environments. |
| Wu et al. (2024) | Advanced inference methods like RE-BASE tree search allow smaller models to match the performance of significantly larger ones. | High-efficiency inference, performance optimization for small models. |
| Shao et al. (2024) | Increasing datastore size in retrieval-augmented models consistently improves performance under the same compute budget, without evident saturation. | Retrieval-augmented language models, knowledge-intensive tasks, compute-efficient architectures. |
| Clark et al. (2022) | Routing-based models show diminishing returns at larger scales, requiring optimal routing strategies for efficiency. | Routing-based models, MoEs, transformer scaling. |
| Krajewski et al. (2024) | Fine-grained MoEs achieve up to 40× compute efficiency gains when expert granularity is optimized. | Mixture of Experts models, large-scale compute efficiency. |
| Frantar et al. (2023) | Sparse model scaling enables predicting optimal sparsity levels for given compute budgets. | Sparse models, structured sparsity optimization, parameter reduction. |

Table 10: Scaling laws of efficient models.

| Paper | Key insights | Applicability |
|---|---|---|
| Clark et al. (2022) | Routing-based models show diminishing returns at larger scales, requiring optimal routing strategies for efficiency. | Routing-based models, MoEs, transformer scaling. |
| Krajewski et al. (2024) | Fine-grained MoEs achieve up to 40× compute efficiency gains when expert granularity is optimized. | Mixture of Experts models, large-scale compute efficiency. |
| Frantar et al. (2023) | Sparse model scaling enables predicting optimal sparsity levels for given compute budgets. | Sparse models, structured sparsity optimization, parameter reduction. |

Table 11: Scaling laws for routing, sparsity, pruning, and quantization.

a student model will perform based on the student model's size, the teacher model's performance and the compute budget allocation between the teacher and the student (Busbridge et al., 2025). We summarize these practical insights in Table 10 for better readability.

## A.6 Data scaling laws

Scaling models involves more than just increasing parameters; optimizing data mixtures, training duration, and vocabulary size also plays a crucial role in enhancing performance and efficiency. Data mixing laws allow AI practitioners to accurately predict optimal data compositions before training, leading to 27% fewer training steps without compromising accuracy (Ye et al., 2024). Techniques like REGMIX optimize data selection using proxy models and regression, reducing compute costs by 90% compared to manual data selection (Liu et al., 2024). Meanwhile, AUTOSCALE revealed that data efficiency depends on model scale, where high-quality data like Wikipedia helps small models but loses effectiveness for larger models, which benefit from diverse datasets like CommonCrawl (Kang et al., 2024). For continual learning, the D-CPT Law provided a theoretical framework for balancing general and domain-specific data, guiding efficient domain

| Paper | Key insights | Applicability |
|---|---|---|
| Ye et al. (2024) | Predicts optimal data compositions before training, reducing compute costs by up to 27% while maintaining performance. | Pre-training optimization, data efficiency improvements. |
| Liu et al. (2024) | REGMIX optimizes data mixtures using proxy models, achieving 90% compute savings. | Compute-efficient training, automated data selection, large-scale models. |
| Allen-Zhu & Li (2024) | Language models can store 2 bits of knowledge per parameter, with knowledge retention dependent on training exposure. | Knowledge encoding, model compression, retrieval-augmented models. |

Table 12: Critical scaling laws for data mixing and knowledge capacity.

adaptation and long-term model updates (Que et al., 2024). Additionally, Chinchilla scaling assumptions were challenged by evidence showing that training models for more epochs on limited data can outperform simply increasing model size (Muennighoff et al., 2023). Repeated data exposure remains stable up to 4 epochs, but returns diminish to zero after around 16 epochs, making longer training a more effective allocation of compute resources. Furthermore, the vocabulary scaling law suggested that as language models grow larger, their optimal vocabulary size should increase according to a power law relationship (Tao et al., 2024). Finally, knowledge capacity scaling laws established that language models store 2 bits of knowledge per parameter, meaning a 7B model can encode 14B bits of knowledge – surpassing English Wikipedia and textbooks combined (Allen-Zhu & Li, 2024). Table 12 summarizes the data scaling laws for developing neural models when data is not available in abundance.

## A.7 Scaling laws for reinforcement learning

Scaling laws in reinforcement learning (RL) and reward model optimization reveal both similarities and differences with generative modeling. Single-agent RL follows power-law scaling with model size and environment interactions, with optimal scaling exponents between 0.4-0.8 across tasks lower than the 0.5 exponent observed in language models (Hilton et al., 2023). RL tasks require orders of magnitude smaller models than generative tasks, correlating with task horizon length, which dictates environment interaction scaling. Task difficulty increases compute needs but does not affect scaling exponents, highlighting horizon length as a key factor in RL scaling efficiency.

In board games like Hex which involves multi-agent RL, Jones (2021) showed that AlphaZero performance follows predictable scaling trends, with compute requirements increasing $7\times$ per board size increment for perfect play and $4\times$ for surpassing random play (Jones, 2021). Neumann & Gros (2023) extended this study to Pentago and ConnectFour, proposing scaling laws which show that player strength scales with network size as $\alpha_N \approx 0.88$, performance with compute as $\alpha_C \approx 0.55$, and optimal network size with compute budget as $\alpha_{\text{opt}} \approx 0.63$ (Neumann & Gros, 2023). Larger multi-agent models exhibit higher sample efficiency, though these trends may not generalize to highly complex games like Chess and Go.

Reward model overoptimization in RLHF follows distinct functional forms: Best-of-$n$ (BoN) reward optimization is governed by $d(\alpha_{\text{bon}} - \beta_{\text{bon}} d)$, whereas RL reward optimization follows $d(\alpha_{\text{RL}} - \beta_{\text{RL}} \log d)$, where $d$ represents KL divergence from the initial policy (Gao et al., 2022). RL requires higher KL divergence than BoN for optimization, and reward model overoptimization scales logarithmically with model size, while policy size has minimal impact. These findings reinforce the importance of balancing compute allocation, environment complexity, and optimization techniques to achieve scalable and efficient RL models.

## A.8 Scaling laws for sparse autoencoders

Recent research has established scaling laws for dictionary learning, providing insights into how latent representations and sparsity impact reconstruction error and computational efficiency. Sparse autoencoders with

| Paper | Category | Task | Architecture | Datasets Used | Model Range | Data Range |
|---|---|---|---|---|---|---|
| Kaplan et al. (2020) | Pre-Training | Language Generation | Decoder-only Transformer | WebText2 | 0M - 1B | 22M - 23B |
| Hoffmann et al. (2022) | Pre-Training | Language Generation | Decoder-only Transformer | MassiveText,Github, C4 | 70M - 16B | 5B - 500B |
| Tay et al. (2022) | Pre-Training , Transfer Learning | Language Generation | Switch, T5 Encoder-Decoder, Funnel, MoS, MLP-mixer, GLU, Lconv, Evolved, Dconv, Performer,Universal, ALBERT | Pretraining: C4, Fine-Tuning: GLUE, SuperGLUE, SQuAD | 173M - 30B | |
| Hu et al. (2024) | Pre-Training | Language Generation | Decoder-only Transformer | Large mixture | 40M - 2B | |
| Caballero et al. (2023) | Pre-Training | Downstream Image Recognition and Language Generation | ViT, Transformers, LSTM | Vision pretrained: JFT-300M, downstream : Birds200, Caltech101, CIFAR-100; Language : Big-Bench | | |
| Hernandez et al. (2021) | Transfer Learning | Code Generation | Decoder-only Transformer | Pre-train: WebText2, CommonCrawl, English Wikipedia, Books; FineTune: Github repos | | |
| Abnar et al. (2021) | Transfer Learning | Image Recognition | ViT, MLP-Mixers, ConvNets | Pre-train: JFT, ImageNet21K | 10M - 10B | |
| Mikami et al. (2021) | Transfer learning | Image Recognition | ConvNets | Syntheic Data | | |
| Zhang et al. (2024) | Transfer Learning | Machine Translation and Language Generation | Decoder-only Transformer | WMT14 English-German (En-De) and WMT19 English-Chinese (En-Zh), CNN/Daily-Mail, ML-SUM | 1B - 16B | 84B - 283B |
| Chen et al. (2024c) | Transfer learning | Language Generation | Decoder-only Transformer | Pre-Train: RedPajama v1, Validation: GitHub,ArXiv,Wikipedia, C4, RedPajama validation sets, ProofPile | 43M - 3B | |
| Lin et al. (2024b) | Transfer learning | Language Generation | Decoder-only Transformer, Encoder-Decoder Transformer, Multilingual, MoE | Fine Tune: WMT19 English-Chinese (En-Zh), Gigaword, FLAN | 100M - 7B | |
| Dettmers & Zettlemoyer (2023) | Quantization Inference | Language Generation | Decoder-only Transformer | The Pile, Lambada, PiQA, HellaSwag, Windogrande | 19M - 176B | |
| Cao et al. (2024) | Quantization Inference | Language Generation | Decoder-only Transformer | WikiText2, SlimPajama, MMLU, Alpaca | 500M - 70B | |
| Kumar et al. (2024) | Quantization Pre-Training, Quantization Inference | Language Generation | Decoder-only Transformer | Dolma V1.7 | 30M - 220M | 1B - 26B |
| Chen et al. (2024a) | Inference | Language Generation | Decoder-only Transformer | MMLU Physics, TruthfulQA, GPQA, Averitec | | |
| Snell et al. (2024) | Inference | Language Generation | Decoder-only Transformer | MATH | | |
| Brown et al. (2024) | Inference | Language Generation | Decoder-only Transformer | GSM8K, MATH, MiniF2F-MATH, CodeContests, SWE-bench lite | 70M - 70B | |
| Wu et al. (2024) | Inference | Language Generation | Decoder-only Transformer | MATH500, GSM8K | 410M - 34B | |
| Sardana et al. (2024) | Inference | Language Generation | Decoder-only Transformer | Jeopardy, MMLU, BIG bench, WikiData, ARC, COPA, PIQA, OpenBook QA, AGI Eval, GSM8k, etc | 150M-6B | 1.5B - 1.2T |
| Clark et al. (2022) | Sparsity | Language Generation | Decoder-only Transformer, MoE | MassiveText | 0 - 200B | 0-130B |
| Frantar et al. (2023) | Sparsity | Language Generation, Image Recognition | Encoder-decoder, ViT | JFT-4B, C4 | 1M - 85M | 0 - 1B |
| Krajewski et al. (2024) | Sparsity | Language generation | Decoder-only Transformer, MoE | C4 | 129M - 3B | 16B - 130B |
| Yun et al. (2024) | Sparsity | Language generation | Decoder-only Transformer, MoE | Slim Pajama | 100M - 730M | 2B - 20B |
| Chen et al. (2024b) | Sparsity | Language generation | Decoder-only Transformer | SlimPajama | 500M - 8B | 0.5B |
| Busbridge et al. (2025) | Distillation | Language generation | Teacher-Student Decoder-only Transformer | C4 | 100M - 12B | 0 - 500B |
| Henighan et al. (2020) | Multimodality | Generative Image Modeling, Video Modeling, Language Generation | Decoder-only Transformer | FCC100M, and various modal datasets | 0.1M-100B | 100M |
| Zhai et al. (2022) | Multimodality | Image Recognition | ViT | ImageNet-21K | 5M - 2B | 1M - 3B |
| Alabdulmohsin et al. (2022) | Multimodality | Image Recognition, Machine Translation | ViT, MLP Mixers, Encoder-decoder, Decoder-only Transformer, Transformer encoder-LSTM decoder | JFT-300M, ImageNet, Birds200, CIFAR100, Caltech101, Big-Bench | 10M-1B | 32M-494M |
| Aghajanyan et al. (2023) | Multimodality | Multimodal Tasks | Decoder-only Transformers | OPT, Common Crawl, LibriSpeech , CommonVoice, VoxPopuli, Spotify Podcast, InCoder, SMILES from Zincand People's Speech | 8M - 30B | 5B - 100B |
| Li et al. (2024a) | Multimodality | Multimodal tasks | ViT, Decoder-only Transformer | CC12M, LAION-400M | 7B - 13B | 1M - 10M |
| Jones (2021) | Multi-agent RL | Hex | AlphaZero with neural networks | | | |
| Neumann & Gros (2023) | Multi-agent RL | Pentago, ConnectFour | AlphaZero with neural networks | | | |
| Gao et al. (2022) | RL | Reward Model training with Best of n or RL | Decoder-only Transformers | | | |
| Hilton et al. (2023) | Single-agent RL | ProcGen Benchmark, 1v1 version of Dota2, toy MNIST | ConvNets, LSTM | | 0M - 10M | |
| Ye et al. (2024) | Data Mixture | Language Generation | Decoder-only Transformer | RedPajama | 70M - 410M | |
| Liu et al. (2024) | Data Mixture | Language Generation | Decoder-only Transformer | Pile | | |
| Kang et al. (2024) | Data Mixture | Language Generation | Decoder-only Transformer , Encoder-Decoder Transformer | RedPajama | | |
| Que et al. (2024) | Data Mixture | Language Generation, Continual Pre-training | Decoder-only Transformer | various mixture of Code, Math, Law, Chemistry, Music, Medical | 0.5B-4B | 0.1B-26B |
| Tao et al. (2024) | Vocabulary | Language Generation | Decoder-only Transformer | SlimPajama | 33M - 3B | 0 - 500B |
| Lindsey et al. (2024) | Sparse Autoencoder | Training Autoencoder | Decoder-only Transformer | | | |
| Gao et al. (2024) | Sparse Autoencoder | Find Interpretable Latents | Decoder-only Transformer | | | |
| Shao et al. (2024) | Retrieval | Language Generation | Decoder-only Transformer | language modelling:RedPajama, S2ORC, Downstream : TriviaQA, NQ, MMLU, MedQA | | |
| Muennighoff et al. (2023) | Pre-Training | Language Generation | Decoder-only transformer | C4 | 10M - 9B | 0 - 900B |
| Allen-Zhu & Li (2024) | Knowledge Capacity | Language Generation | Decoder-only transformer | bioD | | |
| Ma et al. (2024) | Graph Supervised learning | Graph Classification Task | InfoGraph, GraphCL, JOAO, GraphMAE | reddit-threads , ogbg-molhiv,ogbg-molpcba | | |
| Diaz & Madaio (2024) | Criticize | | | | | |
| Sorscher et al. (2023) | Criticize | Image Recognition | ConvNets, ViT | SVHN, CIFAR-10, and ImageNet | | |
| Bahri et al. (2021) | Theoretical | | | | | |
| Bordelon et al. (2024) | Theoretical | | | | | |
| Hutter (2021) | Theoretical | | | | | |
| Lin et al. (2024c) | Theoretical | | | | | |
| Sharma & Kaplan (2020) | Theoretical | | | | | |
| Jin et al. (2023) | Downscaling | | | | | |

Table 13: Details on task, architecture of models and training setup for each paper surveyed.

| Paper | Dependent variable | Scaling variable | Functional form |
|---|---|---|---|
| Kaplan et al. (2020) | Pre-Training Loss | Model Parameters, Compute, Data, Training Steps | $L(N, D) = \left[\left(\frac{N_c}{N}\right)^{\frac{\alpha_N}{\alpha_D}} + \frac{D_c}{D}\right]^{\alpha_D}$ |
| Hoffmann et al. (2022) | Pre-Training Loss | Model Parameters, Data | $L(N, D) = \frac{A}{N^\alpha} + \frac{B}{D^\beta} + E$ |
| Tay et al. (2022) | Performance metric | Compute | $P \propto C^\alpha$ |
| Hu et al. (2024) | Pre-Training Loss | Model Parameters, Data | $L(P, D) = \frac{A}{N^\alpha} + \frac{B}{D^\beta} + E$ |
| Caballero et al. (2023) | Performance metric | Model Parameters, Compute, Data, Input Size, Training Steps | $y = a + (bx^{-c_0}) \prod_{i=1}^{n} \left(1 + \left(\frac{x}{d_i}\right)^{1/f_i}\right)^{-c_i * f_i}$ |
| Hernandez et al. (2021) | Data Transferred | Model Parameters, Fine-tuning Data | $D_t(D_f, N) = k(D_f)^\alpha (N)^\beta$ |
| Abnar et al. (2021) | Downstream Error | Upstream Error | $e_{DS} = k(e_{US})^a + c$ |
| Mikami et al. (2021) | Downstream Error | Pre-training Data | $e_{DS} = aD^{-\alpha} + c$ |
| Zhang et al. (2024) | Downstream Loss | Fine-tuning Data, Data, Model Parameters, PET parameter | $\hat{L}(X, D_f) = A * \frac{1}{X^\alpha} * \frac{1}{D_f^\beta} + E$ |
| Chen et al. (2024c) | Downstream performance | Pre-training Loss, Compute | $L(C) = (\frac{C}{C_N})^\alpha; \; P(L) = w_0 + w1 \cdot L$ |
| Lin et al. (2024b) | Downstream Loss | Data, Fine-tuning Data | $L(D) = \frac{B}{D_t + D^\beta} + E$ |
| Dettmers & Zettlemoyer (2023) | Accurancy | Total Model Bits After Quantization | |
| Cao et al. (2024) | Total parameters | Quantization Ratio | |
| Kumar et al. (2024) | Loss | Data, Model Parameters, Training Precision, Post-train Precision | $L(N, D, P_w, P_a, P_{kv}, P_{post}) = A N_{eff}^{-\alpha} + B D^{-\beta} + E + \delta_{PTQ}$ |
| Chen et al. (2024a) | Optimal LLM Calls | Fraction Of Easy And Difficult Queries | |
| Brown et al. (2024) | Coverage | Number Of Samples | $\log(C) = ak^{-b}$ |
| Wu et al. (2024) | Optimal Compute | Model Parameters | $\log_{10}(C) = 1.19 \log_{10}(N) + 2.03$ |
| Sardana et al. (2024) | Pre-Training Loss | Model Parameters, Data | $L(N, D) = \frac{A}{N^\alpha} + \frac{B}{D^\beta} + E$ |
| Clark et al. (2022) | Loss | Model Parameters, Number Of Experts , Data | $\log(L(N, E)) = a \log N + b \log E + c \log N \cdot \log E + d$ |
| Frantar et al. (2023) | Loss | Sparsity, Model Parameters, Data | $L = (a_S(1 - S)^{b_S} + c_S) \cdot \left(\frac{1}{N}\right)^{b_N} + \left(\frac{a_D}{D}\right)^{b_D} + c$ |
| Krajewski et al. (2024) | Loss | Granularity, Model Parameters, Data | $\mathcal{L}(N, D, G) = c + \left(\frac{g}{G^\gamma} + a\right) \frac{1}{N^\alpha} + \frac{b}{D^\beta}$ |
| Yun et al. (2024) | Loss | Model Parameters, Number Of Experts , Data | $\log L(N, D, E) \triangleq \log\left(\frac{A}{N^\alpha} + \frac{B}{E^\beta} + \frac{C}{D^\gamma} + F\right) + d \log N \log E$ |
| Chen et al. (2024b) | Post-Training Loss | Uncompressed Model Loss, pruned ratio, Model parameters before pruning, Post-training Data | $L(N_0, D, \rho, L_0) = L_0 + \left(\frac{1}{\rho}\right)^\gamma \left(\frac{1}{N_0}\right)^\delta \left(\frac{N_C}{N_0^\alpha} + \frac{D_C}{D^\beta} + E\right)$ |
| Henighan et al. (2020) | Loss | Model Parameters, Compute, Data | $L(x) = Ax^{-\alpha} + B$ |
| Zhai et al. (2022) | Downstream Error | Compute | $E = aC^b + c$ |
| Alabdulmohsin et al. (2022) | Loss | Compute, Model Parameters, Data | $\frac{L_x - L_\infty}{(L_0 - L_x)^\alpha} = \beta x^c$ |
| Aghajanyan et al. (2023) | Loss | Model Parameters, Data | $\mathcal{L}(N, D_i, D_j) = \left[\frac{\mathcal{L}(N, D_i) + \mathcal{L}(N, D_j)}{2}\right] - C_{i,j} + \frac{A_{i,j}}{N^{\alpha_{i,j}}} + \frac{B_{i,j}}{|D_i| + |D_j|^{\beta_{i,j}}}$ |
| Li et al. (2024a) | Loss | Model Parameters, Data | |
| Jones (2021) | Elo | Compute, Board Size | $Elo = \left(m_{boardsize}^{plateau} \cdot boardsize + c^{plateau}\right) \cdot clamp(m_{boardsize}^{incline} \cdot boardsize + m_{flops}^{incline} \cdot \log flop + c^{incline}, 0)$ |
| Neumann & Gros (2023) | Game Score | Model Parameters, Compute | $E_i = \frac{1}{1 + (X_j / X_i)^{\alpha_X}}$ |
| Gao et al. (2022) | Gold Reward model scores | Root Of KL Between Initial Policy And Optimized Policy (d) | $R(d) = d(\alpha - \beta \log d)$ |
| Hilton et al. (2023) | Intrinsic performance | Model Parameters, Environment Interactions | $I^{-\beta} = \left(\frac{N_c}{N}\right)^{\alpha_N} + \left(\frac{E_c}{E}\right)^{\alpha_E}$ |
| Ye et al. (2024) | Loss on domain i | Proportion Of Training Domains | $L_i(r_{1...M}) = c_i + k_i \exp\left(\sum_{j=1}^{M} t_{ij} r_j\right)$ |
| Que et al. (2024) | Validation loss | Model Parameters, Data, Mixture Ratio | $L(N, D, r) = E + \frac{A}{N^\alpha} + \frac{B \cdot r^\eta}{D^\beta} + \frac{C}{(r + \epsilon)^\gamma}$ |
| Tao et al. (2024) | Unigram-Normalised loss | Non-vocabulary Parameter, Vocabulary Parameters, Data | $\mathcal{L}_u = -E + \frac{A_1}{N_{nv}^{\alpha_1}} + \frac{A_2}{N_v^{\alpha_2}} + \frac{B}{D^\beta}$ |
| Lindsey et al. (2024) | Reconstruction error | Compute, Number Of Latents | |
| Gao et al. (2024) | Reconstruction loss | Number Of Latents, Sparsity Level | $L(n, k) = \exp(\alpha + \beta_k \log(k) + \beta_n \log(n) + \gamma \log(k) \log(n)) + \exp(\zeta + \eta \log(k))$ |
| Shao et al. (2024) | Downstream Accuracy | Datastore , Model Parameters, Data, Compute | |
| Muennighoff et al. (2023) | Loss | Data, Model Parameters, Epochs | $L(N, D) = \frac{A}{N'^\alpha} + \frac{B}{D'^\beta} + E$ |
| Busbridge et al. (2025) | Student Loss | Teacher Loss, Student Parameters, Distillation Tokens | $L_S(N_S, D_S, L_T) = L_T + \frac{1}{L_T^{c_0}} \left(1 + \left(\frac{L_T}{L_{S,d_1}}\right)^{1/f_1}\right)^{-c_1/f_1} \left(\frac{A}{N_S^{\beta'}} + \frac{B}{D_S^{\beta'}}\right)^{\gamma'}$ |

Table 14: Scaling law forms proposed in different papers we surveyed.

| Paper | Training code | Analysis code | Github link |
|---|---|---|---|
| Kaplan et al. (2020) | N | N | |
| Hoffmann et al. (2022) | N | N | |
| Hoffmann et al. (2022) | N | N | |
| Hu et al. (2024) | Y | N | Link |
| Caballero et al. (2023) | N | Y | Link |
| Hernandez et al. (2021) | N | N | |
| Abnar et al. (2021) | N | N | |
| Mikami et al. (2021) | N | Y | Link |
| Zhang et al. (2024) | N | N | |
| Chen et al. (2024c) | N | N | |
| Lin et al. (2024b) | N | Y | Link |
| Dettmers & Zettlemoyer (2023) | N | N | |
| Cao et al. (2024) | N | N | |
| Kumar et al. (2024) | N | N | |
| Chen et al. (2024a) | Y | Y | Link |
| Snell et al. (2024) | N | N | |
| Brown et al. (2024) | Y | Y | Link |
| Wu et al. (2024) | Y | N | Link |
| Sardana et al. (2024) | N | N | |
| Clark et al. (2022) | N | Y | Link |
| Frantar et al. (2023) | N | N | |
| Krajewski et al. (2024) | Y | Y | Link |
| Yun et al. (2024) | N | N | |
| Chen et al. (2024b) | N | N | |
| Henighan et al. (2020) | N | N | |
| Zhai et al. (2022) | Y | N | Link |
| Alabdulmohsin et al. (2022) | N | Y | Link |
| Aghajanyan et al. (2023) | N | N | |
| Li et al. (2024a) | N | N | |
| Jones (2021) | Y | Y | Link |
| Neumann & Gros (2023) | Y | Y | Link |
| Gao et al. (2022) | N | N | |
| Hilton et al. (2023) | N | N | |
| Ye et al. (2024) | Y | Y | Link |
| Liu et al. (2024) | Y | Y | Link |
| Kang et al. (2024) | Y | Y | Link |
| Que et al. (2024) | N | N | |
| Tao et al. (2024) | Y | Y | Link |
| Lindsey et al. (2024) | N | N | |
| Gao et al. (2024) | Y | Y | Link |
| Shao et al. (2024) | Y | Y | Link |
| Muennighoff et al. (2023) | Y | Y | Link |
| Allen-Zhu & Li (2024) | N | N | |
| Ma et al. (2024) | Y | N | Link |
| Sorscher et al. (2023) | N | Y | Link |

Table 15: Reproducibility of different neural scaling law papers. Reproducibility status of 45 papers surveyed: 22 (48.9%) provided repositories; 29 (64.4%) did not share training code.

top-$K$ selection follow power-law scaling for reconstruction error (MSE) in terms of the number of latents $n$ and sparsity $k$, though this relationship only holds for small $k$ relative to model dimension (Gao et al., 2024). Larger language models require more latents to maintain the same MSE at a fixed sparsity, reinforcing that latent dimensionality must scale with model size for effective reconstruction. Additionally, MSE follows a power-law relationship with the compute used during training, suggesting that efficient scaling strategies must balance sparsity, latent size, and training compute to minimize error effectively. This is reinforced by Lindsey et al. (2024), showing that feature representations follow predictable scaling trends, where larger models develop richer, more interpretable dictionaries as the number of learned features increases.

## A.9 Scaling laws for graph neural networks

Unlike in computer vision and natural language processing, where larger datasets typically improve generalization, graph self-supervised learning methods fail to exhibit expected scaling behavior and performance fluctuates unpredictably across different data scales (Ma et al., 2024). However, self-supervised learning pretraining loss does scale with more training data, but this improvement does not translate to better downstream performance. The scaling behavior is method-specific, with some approaches like InfoGraph showing more stable scaling than others like GraphCL.

## A.10 Scaling laws in robotics

Recent evidence suggests that robotic learning exhibits scalable regularities akin to language and vision, but with embodiment-specific twists. A large meta-analysis of Robot Foundation Models (RFMs) and LLM-based robotics systems reports consistent power-law improvements as model size, data, and compute scale, with scaling exponents for RFMs comparable to those in vision and in some cases exceeding those observed for language-only tasks; the study further emphasizes the role of task diversity and multimodality in realizing these gains (Sartor & Thompson, 2024). Complementing this global view, an extensive empirical investigation of imitation learning for manipulation establishes that generalization success follows an approximate power law in the *diversity* of training environments and object categories, while exhibiting diminishing returns in the *number* of demonstrations per environment once a modest threshold is reached; prioritizing coverage over repetition yields substantially better zero-shot transfer to unseen scenes and objects (Lin et al., 2024a). Together, these findings point to predictive, albeit still empirically grounded laws for robotics: performance scales with model/data/compute in a manner that is measurable and forecastable, provided that datasets broaden along the axes most relevant to embodiment (scene, object, and interaction diversity). Unlike static-language settings, robotics introduces additional constraints that shape scaling behavior: data collection is expensive and safety-critical, rewards can be sparse, sim-to-real gaps complicate extrapolation, and hardware throughput/latency bound feasible training and deployment. Consequently, practical "robotics scaling laws" should be framed over (i) model and dataset scale, (ii) *diversity* rather than counts alone, and (iii) system constraints (embodiment, safety, and real-time compute), yielding design guidance that balances accuracy, robustness, and cost at scale.

## A.11 Scaling laws for diffusion-based models

Recent studies indicate that diffusion models exhibit recognizable scaling regularities, while also introducing domain-specific nuances distinct from language and standard vision settings. For text-to-image *Diffusion Transformers* (DiT), pre-training loss follows a power-law in compute across wide budgets, enabling forecasts of optimal model size and dataset size under fixed compute and showing that loss trends correlate with downstream generative quality (e.g., FID) (Liang et al., 2024). In *video diffusion*, analogous compute–performance regularities hold only when scale is modeled jointly with *optimization hyperparameters*: learning rate and batch size exert outsized influence, and an extended law that predicts their optima as functions of model/data/compute yields tighter fits and cheaper operating points (e.g., comparable performance at markedly lower inference cost) (Yin et al., 2025). Uniquely for diffusion, a second axis of *inference-time scaling* is operative: beyond the diminishing returns of simply increasing denoising steps, search and verification-based sampling converts additional test-time compute into tangible quality gains, effectively extending the scaling frontier post-training (Ma et al., 2025). Together, these results suggest formulating

"diffusion scaling laws" over *training compute, model/data scale, hyperparameter optima, and inference budget*, rather than training alone to obtain predictive, design-useful guidance for both training-time allocation and test-time compute–quality trade-offs.

# B    Reproducibility of scaling laws papers

The reproducibility status of neural scaling law papers presents a mixed landscape in terms of research transparency. We consolidate and provide the links to github code repositories in the Table 15. Among the 45 surveyed papers proposing scaling laws, 22 papers (48.9%) provided repository links, indicating some level of commitment to open science practices. However, more than half of the papers still lack basic reproducibility elements, with 29 papers (64.4%) not sharing training code and 27 papers (60%) withholding analysis code. This comprehensive survey suggests that while there is a growing trend toward reproducibility in neural scaling law research, there remains substantial room for improvement in establishing standard practices for code sharing and result verification.

