# OpenReview forum: "How to Upscale Neural Networks with Scaling Law?"
_TMLR — Accepted by TMLR_

### Review · Reviewer_gvz6 · 2025-09-10

**Summary Of Contributions:**

The paper is a survey on the topic of scaling law. The paper proposes a taxonomy of the topic: first into data scaling and model scaling, and then further divides model scaling into pre-trianing, post-training and inference. The paper then summarizes the research questions into several categories and discusses the papers based on such categorization.

**Audience:**

Yes

**Audience Explanation:**

The paper is a survey on the topic of scaling law, which is a fundamental observation in modern model design.

**Claims And Evidence:**

Yes

**Claims Explanation:**

The paper is a survey and mostly summarizes results from previous papers.

**Requested Changes:**

1. It seems to me unnecessary to mention the number of papers summarized. Table 2 is quite informative, and maybe the paper/references could also be added to the taxonomy figure (Figure 2).

2. The survey does not provide any empirical figures/statistics on the scaling law from previous papers. I think this is crucial to the survey. On the one hand, we need to show the evidence (the model, the data, and the setting) that is pro-scaling-law, and on the other hand, we need to show how the scaling law could be broken under other scenarios (e.g., the MOE settings).

3. I think two more fields are worth mentioning in the survey. One is about the scaling law for diffusion models, and the other is the scaling law in the field of robotics (imitation learning and etc).

---

> ### Author Response · Authors · 2025-10-17
> **Response to Reviewer gvz6 Comments**
>
> We thank the reviewer for meticulously reviewing the paper. We have addressed your comment in here, with appropriate changes being made to the revised manuscript (highlighted in blue color).
>
> **Add empirical figures/statistics; show pro/anti evidence (e.g., MoE).**
>
> **Response.** We explain why laws diverge by tracing their purposes and assumptions: Kaplan’s single slope across size and data, Hoffmann/Chinchilla’s separable compute-optimal form, Caballero’s BNSL allowing regime changes, and data-constrained laws that value repeated tokens; for data strategies we move from mixture-agnostic to mixture-aware, proxy-optimized (REGMIX/AUTOSCALE), and continual (D-CPT) formulations. This framework resolves conflicting trends by showing how redundancy, exposure, and domain counts alter the effective scaling variable. We then provide empirical contrasts where one law fails and another succeeds.
>
> **Revisions.** We added Figures 4–6: Kaplan vs. BNSL under multi-regime loss, Chinchilla vs. data-constrained under repeated epochs, and dense vs. MoE families with an optimal granularity envelope. Tables 4–6 summarize where each law works or fails.
>
> **Include diffusion and robotics.**
>
> **Response.** We appreciate the reviewer for suggesting these changes. We have broadened the scope of our survey paper with condensed syntheses for diffusion and robotics, including the distinctive inference-time scaling axis in diffusion and diversity-driven scaling in robotics.
>
> **Revisions.** We elaborated scaling laws for diffusion and robotics reflecting recent results and added a roadmap figure with an elaborated caption linking these empirical lessons to practical, sustainable scaling practices.

---

### Review · Reviewer_MDBh · 2025-09-11

**Summary Of Contributions:**

This paper surveys the existing landscape of neural network scaling law research. Specifically, the paper focuses on the difference of scaling laws under different architectures, modalities, training scheme, and compression methods. The strength of the paper lies in the broad range of scaling laws surveyed. The paper clearly categorizes existing scaling law research into different research questions and summarizes the conclusion from these papers.

On the weakness side, the paper does a poor job in providing useful insight. The paper overall plainly listed the scaling law equations from different work, without providing further analysis or verifications. Specifically, when different work show different scaling trends, such as the difference between Equation 6 and 7 under RQ3 or Equation 8-12 under RQ5, the paper fails to provide further analysis on how to understand such difference and whether all these scaling laws are correct or each of them only work under a certain condition. Without these anlaysis, the contribution of this survey is minimal. Furthermore, the paper only listed existing scaling laws, without proposing new scenarios that the current sclaing law may not work and motivate future research directions.

**Audience:**

Yes

**Audience Explanation:**

Scaling law is a hot topic for large models, which is within the interests of TMLR audiances.

**Broader Impact Concerns:**

No broader impact concerns.

**Claims And Evidence:**

Yes

**Claims Explanation:**

As a survey paper, this paper listed the observations from existing work and provide no new claims.

**Requested Changes:**

As mentioned in the summary, the paper should add more analysis on the similarity and difference between the scaling law formulations mentioned under the same research questions. Adding a few figures with the trend line of each sclaing law and real model performance will better help understanding when each scaling law is suitable. Furthermore, besides existing work, proposing new directions where the current scaling law does not work well would be a great contribution.

---

> ### Author Response · Authors · 2025-10-17
> **Response to Reviewer MDBh Comments**
>
> We thank the reviewer for meticulously reviewing the paper. We have addressed your comment in here, with appropriate changes being made to the revised manuscript (highlighted in blue color).
>
> **Analysis of scaling laws**
>
> **Response.** We explain why laws diverge by tracing their purposes and assumptions: Kaplan’s single slope across size and data, Hoffmann/Chinchilla’s separable compute-optimal form, Caballero’s BNSL allowing regime changes, and data-constrained laws that value repeated tokens; for data strategies we move from mixture-agnostic to mixture-aware, proxy-optimized (REGMIX/AUTOSCALE), and continual (D-CPT) formulations. This framework resolves conflicting trends by showing how redundancy, exposure, and domain counts alter the effective scaling variable. We then provide empirical contrasts where one law fails and another succeeds.
>
> **Revisions.** In RQ1 we added the evolution narrative and cross-references to Table 4 and Figures 4–5; in RQ3 we integrated mixture-aware and continual formulations with their domains of validity (Table 5) and connected them to the Figure 5 contrast.
>
> **Future directions**
>
> **Response.** We elaborate section 6 (future directions) with better syntheses and practical guidelines on progressing to the future, scalable and practically-adaptable scaling laws.
>
> **Revisions.** We introduced a "Beyond Scale" section (section 6) proposing quality-weighted tokens, inference budgets, Pareto fronts, and domain-specific laws for future progression of the field.

---

### Review · Reviewer_WG6m · 2025-10-06

**Summary Of Contributions:**

This paper is a survey on existing scaling law methods in various contexts such as language or vision, and various “attributes to scale” such as compute, model size, or data. The methods are organised into a clear taxonomy, and the paper provides recommendations on which method to use for which use case. Beyond simple technical descriptions, the paper also presents advantages and criticism of these methods, as they are presented in various contributions of the literature on scaling laws.

**Audience:**

Yes

**Audience Explanation:**

Some people might be interested in a survey pointing to recent papers in scaling laws. Some practitioners might be interested in the roadmap provided by Figure 4.

**Claims And Evidence:**

No

**Claims Explanation:**

Strengths:

- The paper provides a good and exhaustive summary of recent papers on scaling laws.
- The methods are well organised into a tree structure separating data and model scaling, the different phases of training: pre-training, post-training and inference and the different use cases.
- Figure 4 provides guidelines to map specific use cases to specific scaling law methods.
- The paper acknowledges and describes criticisms of scaling laws existing in the literature.

Weaknesses:

- The major weakness of this paper is the contribution that is very limited for TMLR standards. There are no novel findings, no experiments, no novel insights, everything is just taken from existing paper and the paper feels more like a blog post covering existing literature on scaling laws. Interesting contributions would have been to compare some of these scaling laws in practical setups and provide actual insights based on fair comparisons, comparing different scaling laws in similar setups. The paper also points to the lack of reproducibility in most publications, doing an empirical study would have been a good opportunity to verify the reproducibility of some of the existing approaches.

- I share the doubts and criticism against scaling laws and the possibility to formalize them properly. I do think that a particular scaling law is too specific to a particular setup, which makes it unusable for other setups. In practice, we often have to make our own scaling laws based on intuition and there is no single formula that can really fit our problem. If you take the example of Chinchilla’s scaling laws, they correspond to a very particular mathematical formulation, but the community realised only a few months after they were published that they were wrong and that training smaller LLMs with x10 times more data than recommended was very effective. Nowadays no one uses Chinchilla’s scaling laws in practice and this paper still recommends using them.

- Even without proposing experiments or actual new insights, the paper could be improved a lot, mainly by providing more details and explanations in some of the sections. In section 3, introducing better the concept of optimization for scaling laws, formalizing losses and objectives , providing more details on initialization and validation, with examples. In section 4 providing more intuition behind the presented formulas that all seem to come from nowhere.

**Requested Changes:**

- Proposing concrete experiments to validate, reproduce and compare scaling laws
- Providing way more details, intuitions, explanation for the concepts presented in the paper

---

> ### Author Response · Authors · 2025-10-17
> **Response to Reviewer WG6m Comments**
>
> We thank the reviewer for meticulously reviewing the paper. We have addressed your comment in here, with appropriate changes being made to the revised manuscript (highlighted in blue color).
>
> **Contribution is limited; no new insights/experiments; reads like a blog post.**
>
> **Response.** We appreciate the reviewer for suggesting these changes. The revised manuscript explains why specific law forms arose, how to fit and validate them, and when each succeeds or fails. We complement this analysis with empirical contrasts that make the failure modes concrete (single-slope vs broken; Chinchilla vs data-constrained; dense vs MoE).
>
> **Revisions.** We substantially revised Section 3, adding explicit guidance on optimizer choice (BFGS/L-BFGS, least squares, and when adaptive optimizers are appropriate), loss functions (MSE/MAE/Huber and log-variants), initialization, and validation, plus a limitations paragraph on sensitivity to loss/optimizer/initialization. We added Section 2 intuition for power-law forms and regime breaks. We introduced Tables 4–6 that map laws to effective/ineffective regions and Figs. 4–6 that demonstrate law failure/success on controlled datasets.
>
> **Chinchilla is too specific and became wrong in practice; the paper still recommends it.**
>
> **Response.** We do not present Chinchilla as universally correct, in fact we clarify that it is appropriate for dense LMs on abundant, curated, i.i.d. text when parameters and data are co-optimized, and we identify settings where it mispredicts: data-limited or repeated-exposure regimes, high-redundancy mixtures, and several multimodal scenarios. We contrast it with data-constrained scaling laws that better explain returns when reusing data and often recommend training smaller models longer at fixed budgets.
>
> **Revisions.** Table 4 we elaborate the success and failure scenarios for different model scaling laws, including Chinchilla. Figure 5 provides an empirical case where Chinchilla’s prediction continues to drop while observed test loss saturates; a data-constrained curve matches observations and recommends reallocating compute toward more epochs on smaller models.
>
> **Laws are setup-specific; we end up making our own.**
>
> **Response.** We acknowledge variability across setups and present piecewise/broken laws, mixture-aware laws, and MoE-aware laws as portable templates with explicit knobs -- breakpoints, mixture ratios, and expert granularity, that practitioners can re-fit to their data. We add concrete validation practices so such "own laws" are sound: test for regime breaks, report breakpoint uncertainty, and quantify extrapolation error.
>
> **Revisions.** Figure 4 shows a single-slope fit missing a multi-regime curve while BNSL with learned breakpoints succeeds; Table 4 advises when to favor piecewise fits. Table 5 (mixture-aware, continual D-CPT) and Table 6 (MoE interaction and granularity \(G\)) identify the right knobs and their failure cases.
>
> **Add details to Section 3 (optimization/objectives/initialization/validation) and intuition for formulas.**
>
> **Response.** We added the requested methodological specifics and placed formulas in context, what assumptions yield separable vs. piecewise forms and why slope changes appear at data- or capacity-driven phase transitions.
>
> **Revisions.** Section 3 now enumerates optimizer, loss, and initialization choices with trade-offs and details validation metrics; Section 2 adds mechanistic intuition for power-laws and other methodological concepts.

---

### Author Response · Authors · 2025-09-28
**Inquiry about the pending reviews**

Dear AE,

We are awaiting the response from the pending reviewers (if any). Although we have started working on documenting the rebuttal for reviewers MDBh and gvz6, we are waiting for your formal confirmation before submitting our responses and revised manuscript.

Kindly confirm the timeline by when we can expect the third review and start the rebuttal.

---

### Author Response · Authors · 2025-10-17
**Official Rebuttal of Authors to All Reviewers**

We thank reviewers WG6m, MDBh, and gvz6 for their thoughtful, constructive feedback, which directly shaped our revisions, sharpening methodological rigor, adding clarified intuition, tables and figures, and expanding the scope to diffusion/robotics and practical roadmaps. Their critiques on contribution, conflicting trends, and empirical evidence prompted concrete updates, including an expanded fitting/validation section, law-evolution narratives with domains of validity, and new empirical illustrations and guidance.

All the changes are highlighted in blue color, in the revised manuscript.

---

### Decision · Action_Editor_RsXg · 2025-11-19

**Recommendation:** Accept as is

**Audience:**

Yes

**Audience Explanation:**

I believe a survey on the different papers on scaling laws would be a nice reference for the TMLR audience to use.

**Claims And Evidence:**

Yes

**Claims Explanation:**

The submission provides a comprehensive survey which the authors have expanded with additional explanations and insights after addressing reviewers' concerns, providing where different scaling laws apply and break.